# Influence of AAV vector tropism on long-term expression and Fc-γ receptor binding of an antibody targeting SARS-CoV-2
Jannik T. Wagner [1], Sandra M. Müller-Schmucker[1], Wenjun Wang[2], Philipp Arnold[3], Nadja Uhlig [4], Leila Issmail [4], Valentina Eberlein [4], Dominik Damm[1], Kaveh Roshanbinfar[5], Armin Ensser[1], Friederike Oltmanns[1], Antonia Sophia Peter [1], Vladimir Temchura[1], Silke Schrödel[6], Felix B. Engel [5], Christian Thirion[6], Thomas Grunwald[4], Manfred Wuhrer [2], Dirk Grimm [7] & Klaus Überla [1]✉

Long-acting passive immunization strategies are needed to protect immunosuppressed vulnerable groups from infectious diseases. To further explore this concept for COVID-19, we constructed Adeno-associated viral (AAV) vectors encoding the human variable regions of the SARS-CoV-2 neutralizing antibody, TRES6, fused to murine constant regions. An optimized vector construct was packaged in hepatotropic (AAV8) or myotropic (AAVMYO) AAV capsids and injected intravenously into syngeneic TRIANNI-mice. The highest TRES6 serum concentrations (511 µg/ml) were detected 24 weeks after injection of the myotropic vector particles and mean TRES6 serum concentrations remained above 100 µg/ml for at least one year. Anti-drug antibodies or TRES6-specific T cells were not detectable. After injection of the AAV8 particles, vector mRNA was detected in the liver, while the AAVMYO particles led to high vector mRNA levels in the heart and skeletal muscle. The analysis of the Fc-glycosylation pattern of the TRES6 serum antibodies revealed critical differences between the capsids that coincided with different binding activities to murine Fc-γ-receptors. Concomitantly, the vector-based immune prophylaxis led to protection against SARS-CoV-2 infection in K18-hACE2 mice. High and long-lasting expression levels, absence of anti-drug antibodies and favourable Fc-γ-receptor binding activities warrant further exploration of myotropic AAV vector-based delivery of antibodies and other biologicals.

The outbreak of the SARS-CoV-2 pandemic initiated rapid development of mRNA- and vector-based vaccines[1–3]. These vaccines have a sufficient safety profile, are highly immunogenic and provide protection against severe COVID-19[4–9]. Despite the availability of efficacious vaccines, immunodeficient patients with impaired immune responses to vaccination remain a vulnerable group. Prominent examples are recipients of organ or stem cell transplants and cancer patients under severe immunosuppressive chemotherapy. All these patients could benefit from alternative prophylactic approaches, such as passive immunization[10–14].

During the last decades, Adeno-associated virus (AAV) vectors have been extensively studied for gene therapy of genetic diseases. AAVs are small, non-enveloped viruses and can harbor transgenes up to 5 kb[15–17]. Their advantages for clinical use are low immunogenicity, high stability, and an effective and long-lasting transgene delivery[18–23]. More recently, AAV vectors were also explored for the delivery of genes encoding antiviral antibodies[15,24–27]. Fuchs et al. could show that the delivery of a neutralizing antibody against simian immunodeficiency virus (SIV) to non-human primates could efficiently prevent infection in a SIVmac239 challenge

[1]Institute of Clinical and Molecular Virology, University Hospital Erlangen, Friedrich-Alexander-Universität Erlangen-Nürnberg (FAU), Erlangen, Germany. [2]Center for Proteomics and Metabolomics, Leiden University Medical Center, Leiden, Netherlands. [3]Institute of Functional and Clinical Anatomy, Friedrich-Alexander-Universität Erlangen-Nürnberg (FAU), Erlangen, Germany. [4]Fraunhofer Institute for Cell Therapy and Immunology (IZI), Preclinical Validation, Leipzig, Germany. [5]Experimental Renal and Cardiovascular Research, Department of Nephropathology, Institute of Pathology, Friedrich-Alexander-Universität Erlangen-Nürnberg (FAU), Erlangen, Germany. [6]Sirion Biotech GmbH, Graefelfing, Germany. [7]Department of Infectious Diseases/Virology, Section Viral Vector Technologies, Medical Faculty and Faculty of Engineering Sciences, University of Heidelberg; BioQuant Center, BQ0030, University of Heidelberg; German Center for Infection Research (DZIF), German Center for Cardiovascular Research (DZHK), partner site, Heidelberg, Germany. ✉e-mail: klaus.ueberla@fau.de

model[28,29]. In one of the animals, the AAV vector-encoded antibody persisted for over six years at high serum concentrations[30]. Similarly, Rghei and colleagues were able to induce a long-lasting expression of an mAb targeting Marburg virus in sheep for nearly three years[31]. However, several studies in animals and human volunteers also reported cases of the emergence of anti-drug antibodies (ADA) coinciding with decreasing serum concentrations of the vector-encoded antibodies[29,30,32–35]. The anti-drug antibody response was predominantly directed against the variable region of the neutralizing antibodies and the magnitude of the anti-drug antibody response correlated with the sequence divergence of the delivered antibody from its germline precursor sequence[32]. In addition, the type of the AAV capsid and the organ tropism for expression of the antibody have also been proposed to affect the frequency of emergence of anti-drug antibodies with a higher risk associated with muscle-specific antibody expression[29].

Since SARS-CoV-2 neutralizing antibodies harbored only a few mutations compared to their parental germline sequence[36–38], anti-drug antibodies may be less of a problem for AAV-vectored prophylaxis of COVID-19. In order to study the development of anti-drug antibodies targeting human SARS-CoV-2 antibodies in a syngeneic mouse model, we took advantage of TRIANNI mice. In this transgenic model, the murine immunoglobulin repertoire of the variable regions of heavy and light chains is replaced by the human repertoire (Patent US 2013/0219535 A1). We delivered a monoclonal SARS-CoV-2-targeting antibody that had been raised in TRIANNI mice and showed potent neutralization[36] by packaging the encoding gene in a hepatotropic AAV8 capsid or by a highly myotropic AAV variant (AAVMYO), an engineered capsid variant with high specificity for muscle cells[39]. Both AAV capsids led to high serum levels of the SARS-CoV-2 antibody for at least 52 weeks without any evidence of emergence of anti-drug antibodies. Instead, we observed that the AAV

capsid tropism affected the glycosylation profile of the encoded antibody and the binding to Fc-γ receptors. Moreover, vector-based immune prophylaxis resulted in full protection against lethal SARS-CoV-2 infection in K18-hACE2 mice.

## Results

### Construction and characterization of AAV vectors encoding TRES6

For the initial characterization experiments, an AAV expression vector encoding the TRES6hu antibody, a humanized version of the original neutralizing TRES6 antibody of murine origin[36], was constructed. A monocistronic construct design was chosen to match the limited encapsulation capacity of AAV vectors. The coding sequences of the heavy and light chains of TRES6hu were cloned head-to-tail with a furin cleavage site at the C-terminus of the heavy chain and a self-cleaving 2A peptide of the foot-and-mouth disease virus (F2A) immediately upstream of the light chain, as described previously[40,41] (Fig. 1). CMV enhancer and promoter, a human beta-globin intron in the 5′untranslated region and the woodchuck hepatitis virus posttranscriptional regulatory element (WPRE) were included to increase expression levels (Fig. 1).

Following transient transfection of the AAV vector construct into HEK293T cells, the correct expression of the TRES6hu IgG1 heavy and kappa light chains was analysed by SDS-PAGE and western blot. In comparison to a purified monoclonal TRES6 antibody (mAb), the first vector construct (pAAV-TRES6hu SGSG) expressed light and heavy chains of the expected molecular weight, but additional minor bands of higher molecular weight were also visible (Fig. 2a). The size of the additional 60 kDa band is consistent with insufficient cleavage at the furin cleavage site while the size of the additional 85 kDa band approximates the size of a non-cleaved heavy

**AAV-TRES6hu SGSG/GSGS**

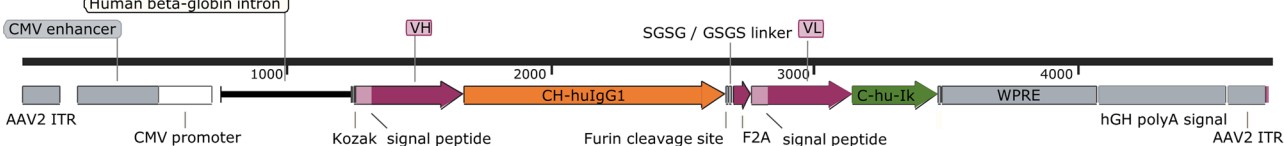

**AAV-TRES6hu LCV**

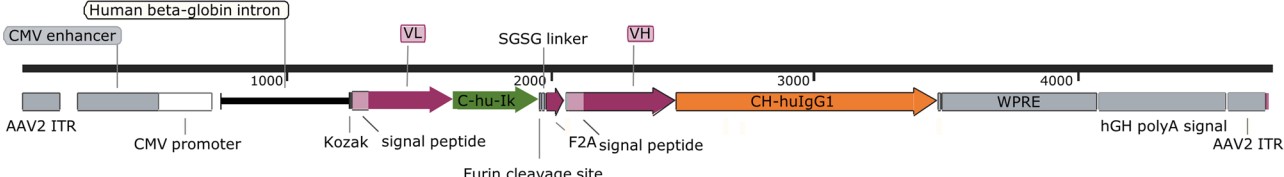

**AAV-TRES6**

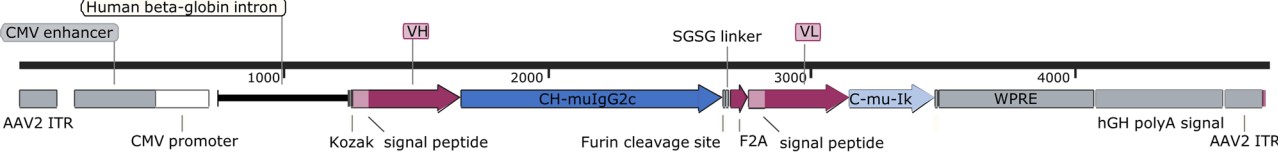

**Fig. 1 | Functional elements of AAV-TRES6 vector constructs.** Depicted are schemes of the AAV-TRES6hu SGSG/GSGS, the AAV-TRES6hu LCV, and the AAV-TRES6 vector constructs. ITR inverted terminal repeat, VH variable region of the immunoglobulin heavy chain of the TRES6 antibody, CH-huIgG1 constant region of human IgG1 heavy chain, CH-muIgG2c constant region of the murine IgG2c heavy chain, SGSG/GSGS serine/glycine linker, F2A F2A self-cleaving peptide of the foot-and-mouth disease virus, VL variable region of the immunoglobulin light chain of TRES6 antibody, C-hu-Igk constant region of the human kappa light chain, C-mu-Igk constant region of the murine kappa light chain, WPRE woodchuck hepatitis virus posttranscriptional regulatory element, hGH polyA signal human growth hormone polyadenylation signal.

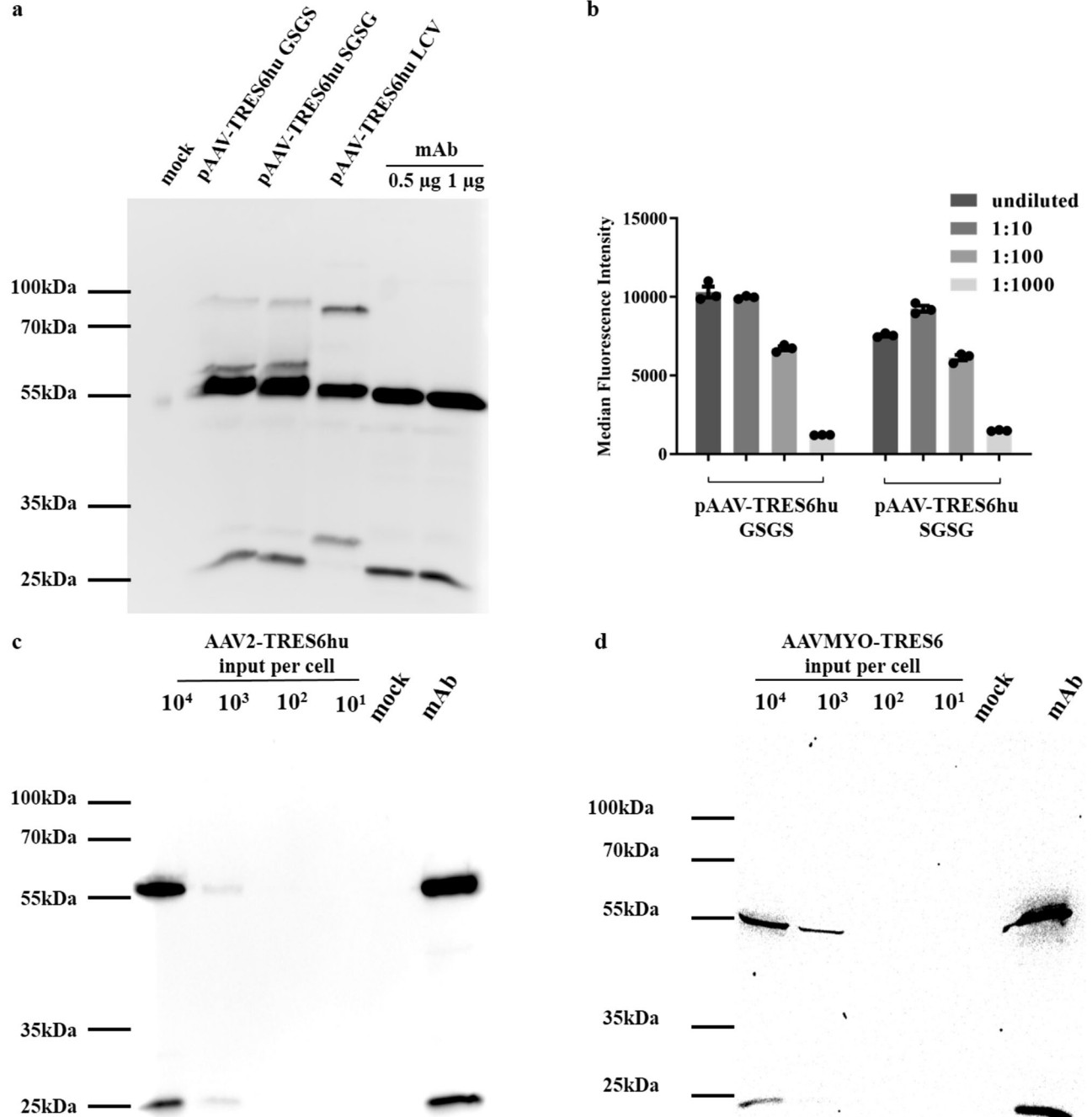

**Fig. 2 | Characterization of AAV vector constructs. a** Western blot analysis of supernatants of HEK293T cells transfected with the indicated AAV vector constructs stained against human IgG1. Supernatants of mock-transfected cells and purified TRES6hu antibody at the indicated amounts were included as negative and positive controls, respectively. **b** Tenfold serial dilutions of supernatants of cells transfected with the indicated AAV vector constructs were analysed by a flow cytometric binding assay for the SARS-CoV-2 spike protein (one representative is shown out of three experiments). **c** Anti-human IgG1 Western blot analysis of supernatants of HEK293T cells transduced with the indicated dose (vg copies per cell) of the AAV vector construct packaged in AAV2 capsids, 0.5 µg of purified TRES6hu antibody served as control. **d** Human iPSC-derived cardiomyocytes were transduced with the AAV-TRES6 vector construct packaged in AAVMYO capsids at the indicated dose. Mock transduced cells and 0.5 µg of purified TRES6 antibody served as controls. All uncropped blots are shown in Fig. S1.

and light chain fusion protein. In an attempt to avoid these additional bands, the original SGSG linker of pAAV-TRES6hu-SGSG was replaced by a GSGS linker (pAAV-TRES6hu GSGS) and the order of the sequences encoding the heavy and light chains were swapped (Fig. 1, pAAV-TRES6hu LCV). The replacement of the serine/glycine linker did not change the bands obtained in the western blot analysis (Fig. 2a). For pAAV-TRES6hu LCV, a properly processed heavy chain could be detected, but the light chain of the expected size of 25 kDa was substituted by a 30 kDa band indicating lack of furin cleavage at the C-terminus of the light chain (Fig. 2a). Despite expression of

some incompletely cleaved immunoglobulin chains, supernatants of HEK293T cells transfected with pAAV-TRES6hu SGSG and pAAV-TRES6hu GSGS seemed to contain high concentrations of antibodies binding to the SARS-CoV-2 spike (S) protein (Fig. 2b). Since the high expression levels obtained by transient transfection of HEK293T cells may overwhelm the furin cleavage capacity of the cells, we also explored correct processing of the encoded antibodies following in vitro AAV transduction. In order to achieve efficient transduction of HEK293T cells, the AAV-TRES6hu-SGSG construct was packaged in AAV2 capsids. Under these

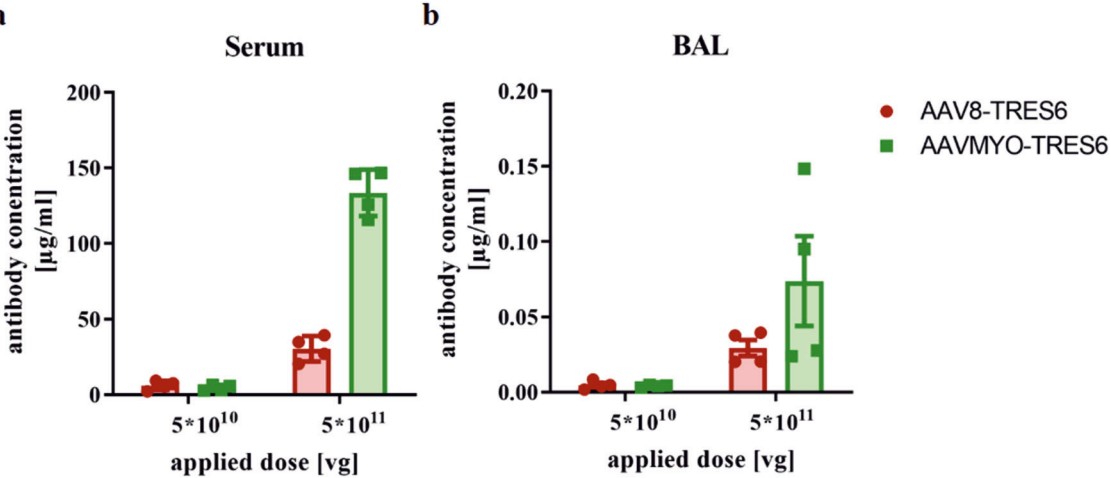

**Fig. 3 | Antibody concentrations in sera and BAL samples.** TRIANNI mice ($n = 4$, ♀ = 2, ♂ = 2) received either AAV8 (red) or AAVMYO (green) particles at the indicated doses of the viral genome (vg) copies. Two weeks after vector application, the mice were sacrificed and antibody concentrations were determined in sera (**a**) and bronchoalveolar lavage (BAL) (**b**) samples.

more physiological expression levels, only correctly processed heavy and light chains were detected (Fig. 2c).

Since the variable regions can be the predominant target of ADA[35], we aimed to explore the longevity of antibody expression and development of ADA in an animal model syngeneic to the variable regions for the human immunoglobulin heavy and light chains. Therefore, TRIANNI mice encoding the repertoire of human VH and VL gene segments fused to the constant regions of the murine immunoglobulin heavy and light chain regions were chosen as recipients of AAV vectors encoding the antibody. The TRIANNI mice had initially been immunized to generate the TRES6 antibody[36] and are, therefore, syngeneic to the VH and VL sequences of TRES6hu. By replacing the human constant regions of the immunoglobulin heavy and light chains of pAAV-TRES6hu SGSG with the constant regions of the murine IgG2c heavy and the murine kappa light chain, the pAAV-TRES6 vector construct (Fig. 1) was cloned encoding a TRES6 antibody that is completely syngeneic to TRIANNI mice reflecting the delivery of human antibodies to humans. This AAV vector construct was packaged into a recently developed myotropic AAV (AAVMYO) capsid[39]. The correct formation of antibody heavy and light chains after the transfer of pAAV-TRES6 with the AAVMYO capsid was confirmed by transducing human-induced pluripotent stem cell-derived cardiomyocytes (hiPSC-CMs). Western blot analysis revealed that the supernatant of the transduced cultures only contained fully processed heavy and light chains at the same molecular size as purified monoclonal TRES6, without the presence of additional, unprocessed products (Fig. 2d). For mouse studies, pAAV-TRES6 was also packaged in a commonly used hepatotropic AAV8 capsid. Imaging of AAV8-TRES6 and AAVMYO-TRES6 particles by negative stain transmission electron microscopy (TEM) revealed that the majority (~98%) of the vector particles were intact, filled, and suitable for in vivo application (Fig. S2).

### In vivo dose finding
The first set of in vivo experiments aimed to identify the optimal dose of vectors to generate a serum antibody concentration of at least 10 µg/ml in TRIANNI mice. For the dose-finding, $5 \times 10^{10}$ viral genomes (vg) or $5 \times 10^{11}$ vg of AAV8-TRES6 or AAVMYO-TRES6 were injected into the lateral tail vein of TRIANNI mice. Fourteen days after injection, the mice were sacrificed, and serum and bronchoalveolar lavage (BAL) fluid was obtained. Analysis of the antibody concentration in the sera by a quantitative ELISA specific for the receptor binding domain (RBD) of the SARS-CoV-2 spike protein revealed that the lower dose failed to exceed the chosen cut-off point of 10 µg/ml for both vectors, whereas the application $5 \times 10^{11}$ vg of AAV8-TRES6 resulted in an average antibody concentration of 30 µg/ml in the sera (Fig. 3a). AAVMYO-TRES6 induced a four-times higher antibody

production than AAV8-TRES6 in TRIANNI mice and resulted in an average antibody serum concentration of 137 µg/ml (Fig. 3a). Measurement of the antibody levels in BAL samples revealed the same trend, with antibody concentrations being ~1000 times lower than in the sera (Fig. 3b). Therefore, a dose of $5 \times 10^{11}$ vg was chosen for all subsequent experiments.

### Efficacy in a murine challenge model
As a proof-of-concept for the vector-mediated immune prophylaxis against SARS-CoV-2 infections, the efficacy of TRES6 antibodies delivered by the AAV8 and AAVMYO capsids was determined in transgenic mice expressing the human angiotensin-converting enzyme 2 (ACE2) under control of the human keratin 18 (K18) promoter. This model allows SARS-CoV-2 infection, replication and disease development[42]. Mice were injected intravenously with $5 \times 10^{11}$ vg AAV8-TRES6 or AAVMYO-TRES6 14 days prior to intranasal challenge with SARS-CoV-2 (300 FFU Wuhan strain). As positive or negative controls, we included groups of mice receiving 3.33 mg/kg recombinant TRES6 or an isotype-matched control antibody five days prior to infection, respectively. Since in vivo expression of antibodies may alter the glycosylation of the antibody depending on the transduced tissue and thus its Fc-dependent effector mechanisms[43], we included a group receiving a recombinant glycosylation-deficient TRES6 N297A mutant also at a dose of 3.33 mg/kg. After the challenge, all mice receiving AAV vector prophylaxis were protected against infection and showed no weight loss and only mild clinical scores, comparable to mice receiving TRES6 antibody prophylaxis (Fig. 4a–c). The viral load in the lungs after AAV-vectored delivery of TRES6 was reduced to a larger extent than in animals receiving recombinant TRES6 (Fig. 4d). This is consistent with the higher antibody concentrations in the sera of mice receiving the AAV vectors (Fig. S3). Interestingly, the N297A mutant of TRES6 was statistically significantly less efficient in reducing viral loads than the parental antibody. Furthermore, one mouse receiving the glycosylation-deficient antibody did not survive the infection. These findings are consistent with previous reports on the role of glycosylation for Fc-effector functions[44].

### In vivo long-term kinetic
A main benefit of AAV-vectored delivery of SARS-CoV-2 neutralizing antibodies compared to recombinant antibodies could be a longer duration of protection. We, therefore, explored the long-term kinetics of TRES6 serum levels after delivery of the AAV vector construct by AAVMYO and AAV8 capsids. Given the potential role of Fc-effector functions observed during the challenge experiment (Fig. 4), we also included an AAV vector construct encoding the N297A mutant of TRES6 deficient in glycosylation of the Fc-fragment. The different AAV vector particles were intravenously

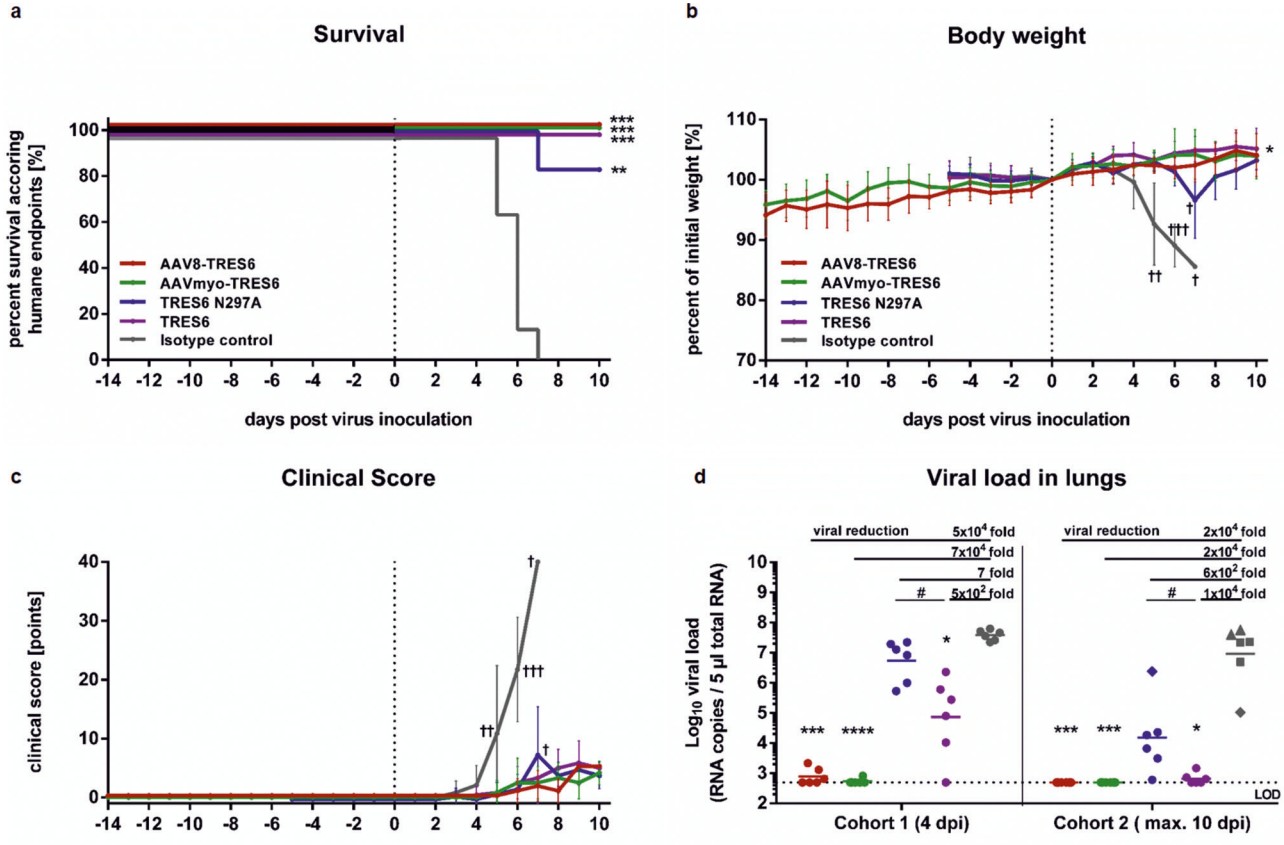

**Fig. 4 | Murine challenge study after AAV-vectored delivery of TRES6.** Female K18 hACE2 mice ($n = 6$ per cohort, ♀ = 6) received intravenously (i.v.) either $5 \times 10^{11}$ vg AAV8-TRES6 (red) or $5 \times 10^{11}$ vg AAVMYO-TRES6 (green) 14 days prior to intranasal SARS-CoV-2 challenge using 300 FFU of Wuhan strain. Control mice received i.v. either 3.33 mg/kg TRES6 (purple), TRES N297A (blue), or isotype control (gray) 5 days prior to the challenge. Cohort 1 ($n = 6$) was euthanized on day 4 and cohort 2 ($n = 6$) was euthanized according to humane endpoints or latest at day 10 after the challenge. The percentage of surviving animals according to humane endpoints are shown (**a**). Statistical evaluation of survival data were performed using the Mantel-Cox test in comparison to isotype control (*$p \leq 0.1$; **$p \leq 0.01$; ***$p \leq 0.001$). Cohort 1 ($n = 6$) was euthanized at day 4. Mice were monitored daily

for body weight (**b**) and cohort 2 ($n = 6$) was euthanized according to clinical score (**c**). Data shown were presented as means ± standard errors (humane endpoints ≥20 points, indicated as †). Viral RNA was extracted from lung homogenates and quantified by SARS-CoV-2-specific RT-qPCR (**d**). Data points shown represent viral copy numbers in each animal with the geometric mean of each group. Each point represents one mouse, whereby circles (●) indicate a survival of four or ten days post-infection and other symbols indicate mice that had to be euthanized according to humane endpoints at day 5 (▲), day 6 (■), or day 7 (♦). Statistical evaluation of the body weight, clinical score, and viral load data were performed by Mann–Whitney $U$-test (*$p \leq 0.05$; ***$p \leq 0.001$; ****$p \leq 0.0001$; not indicated: non-significant). LOD limit of detection, dpi days post-infection.

injected into TRIANNI mice at a dose of $5 \times 10^{11}$ vg. For comparison, the TRES6 antibody was also injected intravenously as a recombinant protein produced ex vivo at a dose of 5 mg/kg body weight. Blood samples were obtained frequently during an observation period of 52 weeks. Four weeks after injection, the mean TRES6 antibody concentrations in the serum of all groups were in the range of 100 to 200 µg/ml (Fig. 5a). Thereafter, a divergent pattern emerged. As expected, the serum concentration of the recombinant TRES6 antibody rapidly declined, with a calculated mean half-life of $6.85 \pm 1.65$ days. The AAV8-encoded TRES6 and TRES6 N297A antibody concentrations remained constant and declined to a concentration of 17.7 µg/ml at 52 weeks after the injection. The N297A mutation seemed to have no major influence on the kinetic of TRES6. In contrast, after delivery of TRES6 by AAVMYO capsids, TRES6 antibody concentrations further increased, reaching a peak of 511 µg/ml at week 24. Even 52 weeks after a single injection, the mean TRES6 serum concentration was still at 138 µg/ml. (Fig. 5a). Differences in the total amount of antibodies produced between weeks 4 and 52 were estimated by an area-under-the-curve analysis and revealed that AAVMYO transduction resulted in 3.7-fold higher TRES6 amounts than AAV8 ($p < 0.0001$), whereas there was no significant difference between the AAV8-TRES6 and AAV-TRES6 N297A groups (Fig. 5b). The antibody concentration in mice lungs showed a similar trend. Here, AAV8-transduced animals showed lower concentrations of antibodies in

the BAL compared to mice from the AAVMYO group at week 4 as well as at week 52 and decreasing concentrations over time. In contrast, AAVMYO-delivered TRES6 showed higher levels in the BAL fluid at the end of the experiment than in week 4 (Fig. 5c).

To confirm the functionality of in vivo expressed antibodies, pseudo-type neutralization assays were performed using lentiviral particles pseudotyped with the spike protein of the Wuhan strain and mouse sera collected four and 52 weeks after vector application. The 50% inhibitory dilution ($ID_{50}$) of week 52 sera from AAVMYO-transduced mice was three-fold higher than in sera from the AAV8 groups. To benchmark these neutralization titers, human sera collected two weeks after the second immunization with the Comirnaty vaccine[45] were also analysed side-by-side with the murine sera. No significant differences in the $ID_{50}$ were observed, indicating that one year after AAV-mediated antibody delivery, serum neutralization activity was comparable to the neutralization activity obtained at the peak of the antibody response induced by two immunizations with a highly effective vaccine (Fig. 5d).

**Anti-drug immune responses and tissue transduction pattern**
Given previous reports on the induction of ADAs after AAV-based delivery of antibodies in NHPs and humans[34,46,47], we performed a bridging ELISA. All samples analysed were comparable to naïve sera and remained below the

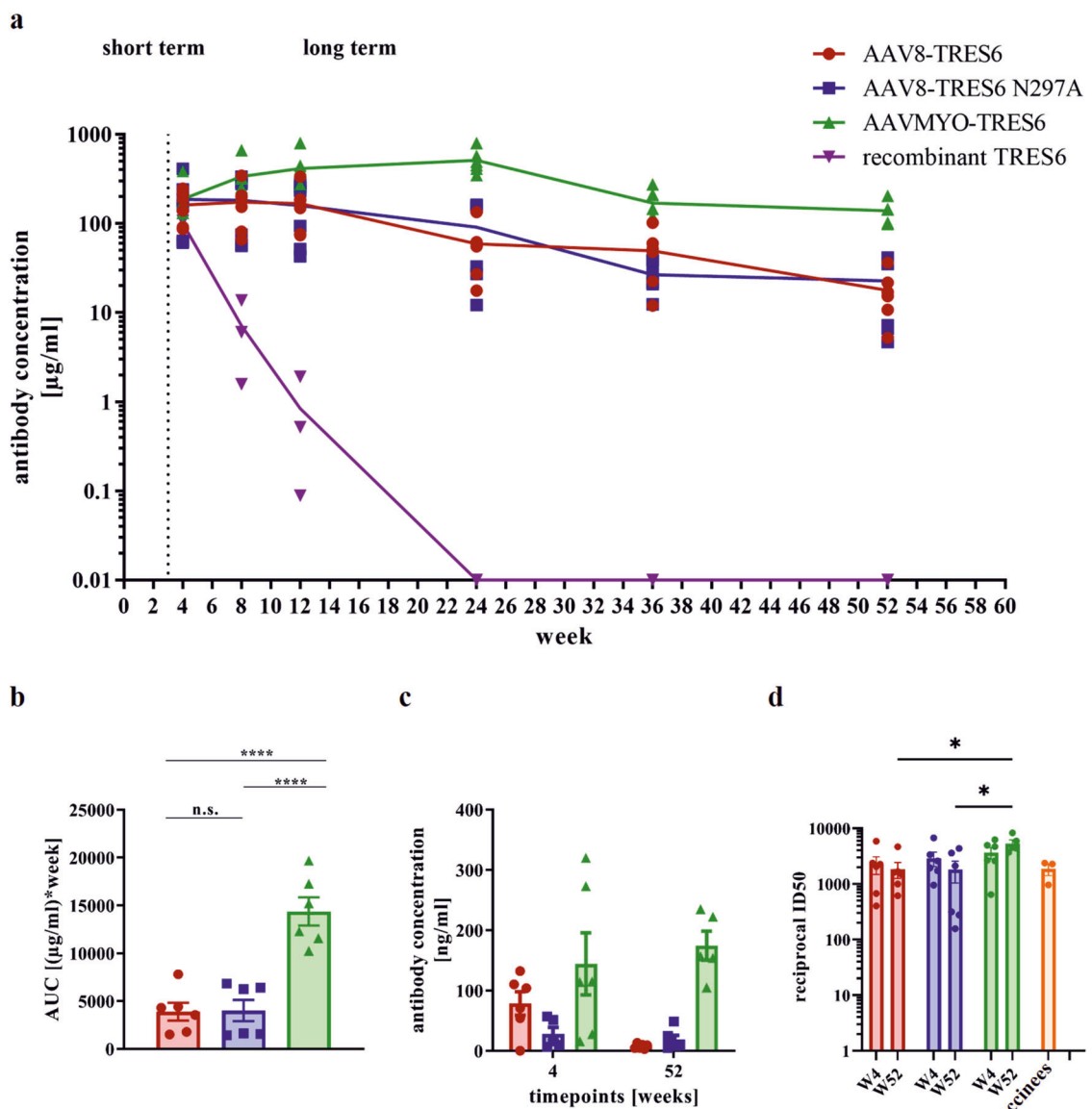

**Fig. 5 | Long-term kinetic of TRES6 antibodies.** TRIANNI mice received intravenously either $5 \times 10^{11}$ vg copies of AAV-vector particles ($n = 6$, ♀ = 3, ♂ = 3) or as control 5 mg/kg body weight TRES6 antibody ($n = 3$, ♀ = 2, ♂ = 1). **a** Each data point represents the mean serum antibody concentration in each group at the indicated time point ± SEM. There was no detectable antibody concentration in the control mice measurable from week 24 onwards. **b** Differences in the total amount of serum antibodies were quantified for vector-transduced mice by calculating the area-under-the-curve (AUC) between weeks 4 and 52 of the long-term kinetic for each animal. **c** At weeks 4 and 52, mice were sacrificed and the TRES6 concentration was determined in the bronchoalveolar lavage fluid of AAV-transduced mice by ELISA. Each bar shows the mean antibody concentration of the group ± SEM. **d** Sera from vector-treated mice that were obtained at weeks 4 and 52 were tested for neutralization in a pseudotype neutralization assay and were compared to sera from human vaccinees. The reciprocal of the dilution at which 50% inhibition is reached (reciprocal $ID_{50}$) is shown per mouse. The bars indicate the group mean value ± SEM. **b, d** Statistical evaluation of the data were performed by Kruskal–Wallis test (one-way ANOVA) and Dunn's pairwise multiple comparison procedures as post hoc test (*$p \leq 0.05$; ****$p \leq 0.0001$; not indicated: $p > 0.05$).

sample cut-off, indicating the absence of anti-drug antibodies (Fig. 6a). For the detection of potential T cell responses to TRES6, splenocytes were stimulated with peptides of TRES6 and its precursor proteins that had the highest immunogenic potential due to the divergence of the TRES6 antibody from the germline sequence. These included peptides from the complementarity-determining regions of TRES6 and the linker sequences between the heavy and light chains of TRES6 (see Fig. 1). Neither CD4- nor CD8-T cell responses could be detected (Fig. S4).

To explore whether the higher antibody concentrations in the AAV-MYO group are due to prolonged circulation of AAVMYO vector particles, single-stranded AAV vector DNA copy numbers were determined by extracting DNA from the sera of the animals at weeks 2, 4, and 8. Since our recombinant AAV vectors have a predominantly single-stranded DNA genome, we additionally treated the purified DNA with a double-strand-specific DNase to ensure that only single-stranded vector DNA is detected. The mean copy numbers of extracted DNA resistant to the double-strand-specific DNase were below 100 copies/μl serum with a decreasing trend over time. No significant differences were evident between the AAV8 and AAVMYO groups (Fig. 6b). Considering the intravenous inoculation dose of $5 \times 10^{11}$ vector DNA copies, these low AAV vector DNA concentrations are unlikely to contribute substantially to the overall number of cells transduced by the injected AAV vector particles.

To confirm the reported tropism of AAV8 for the liver and AAVMYO for skeletal muscles and the heart, RNA was extracted from different organs at weeks 4 and 52. Quantitative RT-PCR revealed expression of the AAV8-delivered vector RNA in the liver, while AAVMYO-delivered vector RNA could only be detected in the skeletal muscle samples and the heart (Fig. 6c).

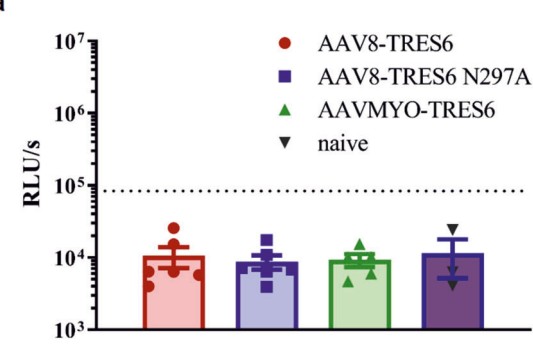

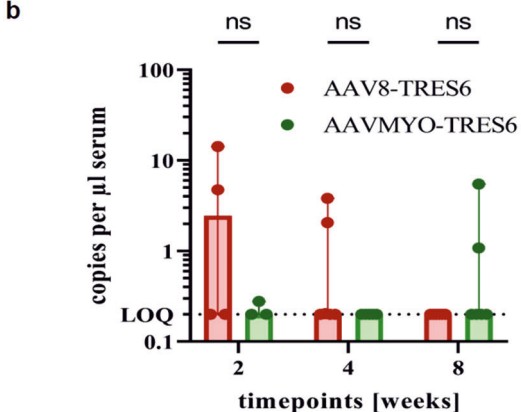

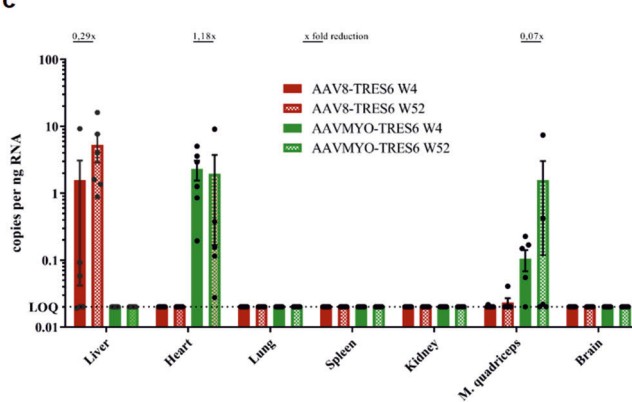

**Fig. 6 | Anti-drug antibody response, serum AAV vector DNA levels, and RNA expression levels. a** Anti-drug antibody levels in TRIANNI mice 52 weeks after injection of the indicated AAV vector particles. The dotted line represents the assay´s sample cut-off, calculated as the sum of the mean of the values from naïve mice and twice their standard deviation. **b** Double-stranded DNase-resistant AAV vector DNA copy numbers per µl DNA extracted from murine sera obtained in weeks 2, 4, and 8 of the long-term kinetic experiment with the indicated AAV vector (group size $n = 6$, ♀ = 3, ♂ = 3). Each bar represents the mean value of each group ± range. For statistical evaluation, a two-way-ANOVA was performed (n.s.: $p > 0.05$). **c** AAV vector RNA copy numbers per ng total RNA extracted from the indicated organs at four and 52 weeks after injection of the indicated AAV vector particles ($n = 6$, ♀ = 3, ♂ = 3). Each bar represents the mean value of each group ± SEM. The fold reduction between the two time points is indicated when the time points are above the lower limit of quantification (LOQ, dotted line).

Interestingly, vector RNA levels do not seem to decline between week 4 and week 52.

### Glycosylation pattern and Fc-γ receptor binding
Given the clear-cut differences in the organ distribution of vector mRNA levels after transfer of the same vector construct with AAV8 or AAVMYO

capsids (Fig. 6c), we concluded that the TRES6 antibodies detected in sera from the AAV8-TRES6 group are predominantly produced in the liver, while the TRES6 antibodies from the AAVMYO group are predominantly derived from skeletal muscles or the heart. Differences in the glycosylation activities of these organs could lead to differences in the glycosylation pattern of the TRES6 antibodies encoded by the different AAV vector particles. To determine the glycosylation pattern of in vivo expressed TRES6 antibodies at position N297, which is the main Fc-glycosylation site, the antibodies were first purified from the mouse sera by spike affinity chromatography and then analysed by mass spectrometry.

The AAV8 sera contained several major non-sialylated glycoforms of TRES6, both with and without fucosylation (Fig. 7a). The profiles of week 4 and week 12 were rather similar. In addition, three N-glycolylneuraminic acid-containing glycoforms were detected, all of them lacking fucose (Fig. S5). Of note, three of the mice (mice BT57, BT75, and BT76) showed high levels of afucosylated glycoforms, whilst the other three mice showed high levels of fucosylated glycoforms. We noticed that the three animals sharing the low fucose phenotype were all male mice, whilst the highly fucosylated antibodies were found in female mice (Fig. S5). The AAVMYO samples showed a more homogenous pattern and were dominated by fucosylated glycan species, with intermediate levels of galactosylation (Fig. 7a). Remarkably, the agalactosylated species H3N4F1 (H: hexose; N: *N*-acetylhexosamine; F: fucose) consistently increased in relative abundance between week 4 and week 12 (Fig. S4). Significant amounts of monoantennary species, as well as oligomannosidic species (Man5; H5N2), were observed. In contrast, overall low levels of sialylation were detected (Fig. 7a and Fig. S6).

Due to these differences in the Fc-glycosylation, we also analysed whether this affects the interaction with the murine Fc-γ receptors. Therefore, the sera from week 8 and week 52 were diluted to a TRES6 concentration of 100 ng/ml, and we then determined the binding of TRES6 to Fc-γ receptors I, IIb, III, and IV. Regarding the binding to Fc-γ RI, AAVMYO-delivered TRES6 showed statistically significant higher binding at week 8 compared to AAV8-TRES6 and AAV8-TRES6 N297A sera (Fig. 7b). At week 52, no differences in Fc-γ RI binding between the two vectors could be seen, whereas the binding was significantly higher compared to the antibody bearing the N297A mutation after AAV8 application (Fig. 7b). At week 8, the sera of AAVMYO-transduced animals showed lower binding to the inhibitory Fc-γ RIIb than the AAV8 group. A similar trend was seen at week 52, but this difference was not statistically significant (Fig. 7c). Binding of Fc-γ RIII did not appear to be affected by the glycosylation pattern at residue N297, consistent with the high binding activity of the N297A mutant (Fig. 7d). Likewise, at week 8, there was no significant difference between the AAV8 and AAVMYO groups in binding to Fc-γ RIV. Nevertheless, at week 52, AAV8-delivered TRES6 antibodies showed statistically significant higher binding than AAVMYO- as well as AAV8-delivered TRES6 N297A antibodies (Fig. 7e). Comparing the changes between week 8 and week 52 within the groups, AAV8-encoded TRES6 showed an increase in binding to Fc-γ RI, RIV and a decreased Fc-γ RIIb interaction (Fig. S7). No significant differences in the Fc-γ R interactions were observed between the two time points in the AAVMYO group (Fig. S7).

### Discussion
In the last years, the experimental use of AAV vectors has been expanded from the treatment of gene deficiency-related diseases to the delivery of neutralizing antibodies against infectious diseases. AAV-delivered mAb targeting the HIV or SIV envelope surface protein reached high antibody concentrations and were able to suppress viremia in challenge experiments. In these experiments, AAV capsids of the subtypes 1, 8, or 9 were investigated[48,49].

In the current study, we performed a side-by-side comparison of AAV vector-based delivery of the same SARS-CoV-2 antibody by a recently developed myotropic AAV capsid[39] and the hepatotropic AAV8 capsid. Clearly, the myotropic AAV capsid led to higher expression in mice

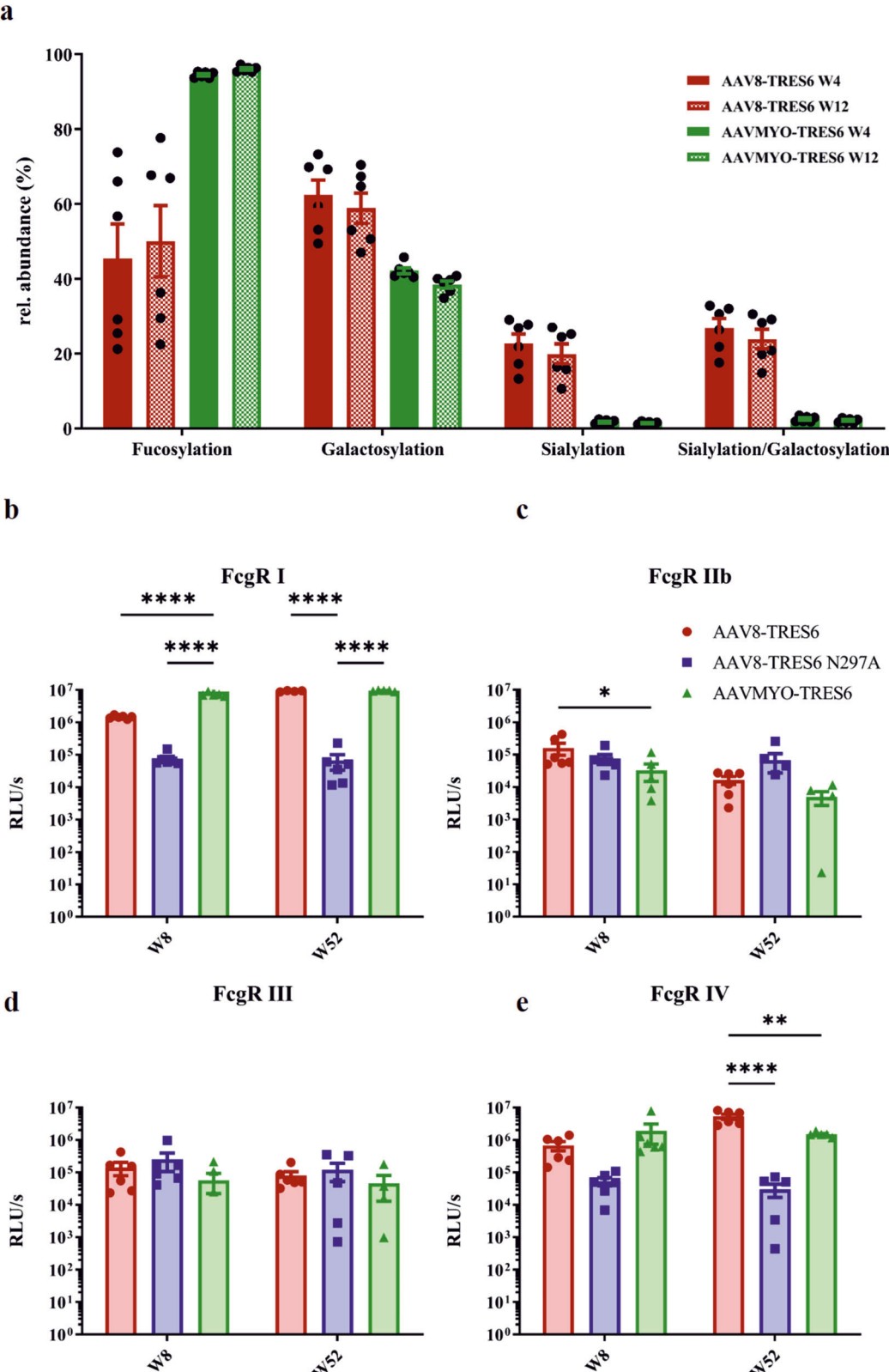

**Fig. 7 | Fc glycosylation and impact on Fc-γ receptor interaction. a** The composition of glycans attached to the amino acid N297 of the antibody Fc part was measured by mass spectrometry for AAV8 (red)- and AAVMYO (green)-delivered TRES6. The samples were obtained in week 4 (W4, filled) and week 12 (W12, shaded) during the long-term kinetic experiment. The binding of in vivo produced TRES6 (AAV8: red; AAVMYO: green) or TRE6 N297A (blue) to the murine Fc-γ receptors RI (**b**), RIIb (**c**), RIII (**d**), and RIV (**e**) was analysed. Sera from week 8 and week 52 during the long-term kinetic experiment were used and diluted to a TRES6 concentration of 100 ng/ml. Each bar shows the mean value of the group ± SEM. Statistical evaluation of the data were performed by Kruskal–Wallis test (one-way ANOVA) and Dunn's pairwise multiple comparison procedures as post hoc test ($*p \leq 0.05$; $***p \leq 0.001$; $****p \leq 0.0001$; not indicated: non-significant).

throughout the entire observation period of 52 weeks. Peak serum concentrations of the SARS-CoV-2 antibody of 511 µg/ml indicate that the AAVMYO-delivered antibody could reach serum levels in an estimated range of 5% of the total IgG concentration in human sera. In terms of safety, the animals did not show any signs of disease. Similarly, i.v. injection of TRES6 into rhesus monkeys resulted in a serum concentration of 100 µg/ml and did not reveal any obvious short-term side effects[50]. Furthermore, the infusion of an anti-HIV mAb reaching median peak antibody concentrations of more than 500 µg/ml was safe and well tolerated in humans[51]. Thus, even such a high concentration of exogenous antibodies seems to be well tolerated. However, it is not possible to exclude that ectopic expression of antibodies may increase the risk of autoimmune responses against the targeted organ and it remains to be investigated if such a potential risk depends on the glycosylation pattern of the antibody. Cases of myocarditis in AAV gene therapy trials[52] suggest a note of caution particularly for AAV vectors with a tropism for cardiomyocytes. Since the transgene seems to play a dominant role in the pathogenesis of myocarditis (reviewed in ref. 53), extensive safety studies are necessary for each vector-encoded antibody.

At the end of our observation period, sera of AAVMYO-transduced mice showed a higher neutralization capacity than the AAV8 groups and a comparable $ID_{50}$ to human sera obtained two weeks after the second vaccination with Comirnaty. This long-term maintenance of antibody levels is in line with a recent study that was able to induce serum antibody concentrations of 128.6 µg/ml by vectored delivery of anti-SARS-CoV-2 antibodies using an AAV-DJ capsid[54].

One limiting factor for the long-term persistence of AAV-delivered antibodies is the formation of a humoral or cellular immune response of the host. In multiple studies in non-human primates or even humans, the induction of anti-drug antibodies led to clearance of the therapeutic antibody from the circulation[28,32–34]. In a Phase-I clinical study, the delivery of the anti-HIV mAb PG9 using AAV1 capsids resulted in a strong ADA response, explaining the absence of detectable levels of PG9. The authors postulated that the unusually long CDR3 loop was responsible for this high immunogenicity[34]. Furthermore, the majority of the study participants showed a T cell response against the AAV1 capsid, that could eliminate the antibody-producing cells[34]. Similar effects were seen when using the AAV8 capsid[33]. The humoral immune response in this study targeted the Fab region of the delivered VRC07 antibody, which also carries long CDRs[33]. In our mouse study, we did not observe any evidence of ADA or T cell response, although the transferred antibody was expressed in muscle tissue that was previously reported to have a high immunogenic potential[47]. The high sequence identity of the TRES6 antibody to its germline precursor sequences (variable region heavy chain: 94.44%; variable region light chain: 97.85%) may explain the low immunogenic potential and long-term expression we observed[36].

The protective potential of vectored immunoprophylaxis using recombinant AAV has already been demonstrated for different infectious agents like SIV or Ebola Virus[25,28]. In terms of SARS-CoV-2 protection, AAV-delivered neutralizing antibodies or soluble decoy receptors were able to prevent severe SARS-CoV-2 infections in hACE2 K18 mice[54,55]. In our comparative study, AAV8- and AAVMYO-transduced mice showed the ability to withstand SARS-CoV-2 infection in terms of survival, clinical score, and weight loss, which is in line with a previous report from Esmagambetov et al.[54]. Already four days after infection, the viral load in the lung of challenged mice was reduced nearly to background levels. Here, differences in mice receiving recombinant TRES6 antibodies could be observed. While the wild-type antibody was able to reduce the viral load, the TRES6 variant carrying a glycosylation-canceling mutation (TRES6 N297A) failed to do so. Furthermore, one mouse receiving TRES6 N297A antibodies had to be euthanized due to the severity of the infection. This clearly shows the already reported influence of antibody glycosylation during viral infections on viral clearance and protection from disease[44].

The analysis of vector RNA levels in different organs revealed a stringent tissue restriction. The AAV8-delivered vector was mainly expressed in hepatic tissue, while AAVMYO delivery led to high vector RNA copy numbers in the heart and the quadriceps muscle. In addition, we observed that RNA copy numbers were higher at week 52 than at week 4 in AAV8-transduced liver and AAVMYO-transduced skeletal muscle tissue. Based on peak antibody serum concentrations at week 12 or 24, vector RNA levels should also peak at some time point between week 4 and week 52. Therefore, higher vector RNA levels until week 52 than at week 4, do not necessarily indicate a continuous increase of vector RNA levels until week 52.

The glycosylation profile of proteins is strongly dependent on the type of producer cell[43]. We now observed that the different cell specificities of AAV8 and AAVMYO capsids indeed resulted in differences in N-glycosylation at asparagine 297 of the vector-encoded antibody Fc-region. The differences in the glycosylation profile coincided with different binding activities to Fc-γ receptors. One striking difference was found in the abundance of the core fucose. Here, AAVMYO-delivered antibodies showed high levels of fucosylation, whereas AAV8 transduction resulted in lower fucosylation with a strong dependency on the sex of mice. All female mice had higher fucose levels than male ones. An analysis of transcriptomic data of liver-derived RNA revealed that in female mice, some enzymes responsible for N-linked glycosylation were elevated in comparison to male samples. This also includes the fucosyltransferase 8 (FUT8, Table S1), which is responsible for IgG fucosylation in humans[56,57]. The importance of this finding lies in the function of afucosylated antibodies. Binding to the matching Fc-γ receptor (mouse: Fc-γ RIV, human: Fc-γ RIIIb) is a crucial trigger for a potent antibody-dependent cellular cytotoxicity (ADCC) response by mononuclear cells of the immune system and is tightly regulated by the presence of core fucose[58,59]. During the response against certain viruses, e.g., HIV, induction of a strong ADCC is expected to be beneficial for proper clearance of infected cells. In this case, the shRNA-mediated knock-down of FUT8 in vitro led to the production of anti-HIV antibodies lacking core fucose and showing increased ADCC[60]. The co-delivery of this shRNA together with the antibody sequence by AAV transduction could lead to an in vivo glycoengineering of produced antibodies towards a strong ADCC response[60]. Nevertheless, during other infections, an improved ADCC activity could be disadvantageous or even harmful. In patients undergoing a SARS-CoV-2 infection, it was found that a strong Fc-γ RIII activation could lead to a severe course of infection[61]. In patients with acute lung failure, elevated levels of afucosylated IgG were found compared to patients with a mild course of infection. High abundance of afucosylated antibodies could lead to hyperinflammation by triggering the Fc-γ RIII cascade and ADCC-mediated tissue damage[61]. Considering this, we conclude that antibodies that were produced following AAVMYO transduction could have a favorable safety profile based on their unanimous high fucose abundance and lower Fc-γ RIV binding. In contrast to AAV8-encoded antibodies, the glycosylation profile of antibodies expressed by AAVMYO was not affected by sex. This is an important point to consider for human applications since differences in fucosylation levels have been reported between men and women[62].

Comparing the application of AAV8 and AAVMYO for the delivery of neutralizing antibodies against SARS-CoV-2, the myotropic AAV capsid seems to be superior. The AAVMYO-transduced mice showed higher antibody titers over the entire observation period of one year. This increased efficiency offers the possibility to reduce the applied AAVMYO dose and still achieve a sufficient antibody titer, resulting in advantages in terms of vector production and patient safety. Furthermore, the AAVMYO-encoded antibodies were highly fucosylated and are, therefore, expected to carry a lower risk of inducing a disadvantageous ADCC mediated by CD16a. However, these hypothetical advantages need to be confirmed experimentally. At the same time, there is a number of different options for further improving AAV-vectored delivery of antiviral antibodies. On the one hand, the required vector dose could be reduced, or the antibody concentrations could be further increased by the introduction of half-life-extending mutations in the Fc-part, like the YTE mutations in human IgG1 antibodies[63]. The use of improved AAVMYO variants such as AAVMYO2/3, which show further liver-detargeting, could lower hepatotoxicity

concerns[64]. Muscle-specific promoters could further improve tissue specificity. On the other hand, the rise of different variants of concern (VOC) of SARS-CoV-2 over the last years limited the effectiveness of the TRES6 antibody[65]. It was shown that mutations that arose in the Delta variant led to escape from neutralization by TRES6. Since the Delta-VOC shares the L452T and T478K mutations with the Omicron-VOC, a loss of neutralization can also be assumed[66]. Accordingly, a broadly neutralizing antibody that could neutralize all VOCs of SARS-CoV-2 would represent a significant improvement. Considering all these points and the findings of this study, the utilization of a myocyte-directed delivery of neutralizing antibodies using AAV vectors could be a useful expansion of the toolbox in the fight against infectious diseases.

## Methods

### Breeding and housing of TRIANNI mice
TRIANNI mice (Patent US 2013/0219535 A1) were used in this study as the primary small animal model. The antibodies in this transgenic mouse strain, which is based on a C57BL/6 background, are composed of human immunoglobulin variable regions and murine immunoglobulin constant regions. The animals were bred and housed in ventilated cages at the Franz-Penzoldt animal facility center of the Friedrich-Alexander-Universität Erlangen-Nürnberg (FAU) and handled under standardized conditions as requested by the Federation of European Laboratory Animal Science Association. All practical steps were approved by the Government of Lower Franconia (license 55.2.2-2532-2-1344).

Since male TRIANNI mice homozygous for the human Ig heavy chain (HH) are infertile, breeding pairs were composed of heavy chain homozygous females and heterozygous males (HH × Hh). The genotype of the offspring in the Ig heavy chain locus as well as the kappa and lambda light chain loci was evaluated via PCR before the mice were assigned for experiments.

### Generation of AAV vector constructs
The initial TRES6hu IgG1-encoding AAV vector construct (pAAV-TRES6hu) was synthesized and cloned in a custom expression vector by GeneArt (Thermo Fisher Scientific, Waltham, MA, USA). The sequence of the TRES6hu vector construct has been deposited at GeneBank (accession number PP740717). For the construct harboring the GSGS-interchain linker, a gene segment (5′-caaccactacacgcagaagagcctctccctgtctccgggtaaacgga agagaagaggatctggctctgccctgtgaagcagaccctgaacttcgatctgctgaagctcgccggcgac gtg-3′) was ordered in a pMA-T vector backbone from GeneArt (Thermo Fisher Scientific). The linker sequence was inserted into pAAV-TRES6hu by digestion with *Sap*I and *Mre*I restriction enzymes and ligated using T4 ligase (all Thermo Fisher Scientific).

To generate pAAV-TRES6hu LCV, the order of the TRES6hu antibody heavy and light chains was swapped. The heavy constant and variable regions of the pAAV-TRES6hu plasmid were amplified with the forward primer 5′-aagctcgccggcgacgtggaaagcaatcctggacctatggagtttgggctgagctggg-3′ and reverse primer 5′-gaggttgattgcgcggccgctcatttacccggagacagggagagg-3′ using the Q5® High-Fidelity DNA polymerase system (New England Biolabs GmbH, Ipswich, MA, USA) following the manufacturer's protocol. The amplicon was inserted into the pAAV-TRES6hu plasmid using *Mre*I and *Not*I restriction enzymes (all Thermo Fisher Scientific) resulting in an intermediate plasmid containing two heavy chain elements (pAAV-TRES6hu-2xHC). Nucleotide sequences of the constant and variable region of the immunoglobulin light chain were then amplified from the pAAV-TRES6hu plasmid using the Q5 PCR kit (New England BioLabs GmbH) following the manufacturer's protocol using the forward primer 5′-gcgactcgaggccgccaccatggacatgagggtccctgc-3′ and reverse primer 5′-gatcc agatcttctttccgacactctcccctgttgaag-3′. The amplicon was inserted between the *Xho*I and *Bgl*II sites of the pAAV-TRES6hu-2xHC plasmid using the respective restriction enzymes (New England BioLabs GmbH). The sequence of the pAAV-TRES6hu-LCV has been deposited at GenBank (accession number PP740718).

For the mouse experiments, pAAV-TRES6 was constructed by amplifying the constant region of the murine kappa light chain of the TRES224 antibody[36] with the forward primer 5′-gaccaagcttgagat caaacgggctgatgctgcaccaac-3′ and reverse primer 5′-ggcggccgcctaaca ctcattcctgttg-3′ with the Q5 PCR kit (New England Biolabs GmbH) following the manufacturer's protocol and insertion of the amplicon via *Hind*III and *Not*I digestion (New England Biolabs GmbH) into the pAAV-TRES6hu plasmid creating the pAAV-TRES6hu-mk plasmid. The murine constant heavy chain was amplified from a plasmid encoding the IgG2c constant heavy chain (GenBank Sequence ID: LC037230.1, nucleotide position 496–1503) with the forward primer 5′-gtctggggccaagggaccacggtcac cgtctcctcaagcgctaaacaacagcccatcggtcta-3′ and the reverse primer 5′-ctgctt cacaggggcgccagatccagatcttctcttccgtttacccagagaccgggagatggtcttagtcg-3′. The amplicon was inserted by Gibson Assembly after restriction digestion of the pAAV-TRES6hu-mk plasmid with *Psh*AI and *Bgl*II (New England BioLabs GmbH) using the Gibson Assembly Master Mix (New England BioLabs GmbH) following the manufacturer's protocol. The N297A mutation had initially been introduced into a plasmid encoding the murine b12 antibody by mutagenesis PCR (forward primer: 5′-acctggaactctggatccctgtccagtggt gtgca-3′, reverse primer: 5′-ctgaccacccggagagtactggcgtaatcctctctatgggtt-3′). The IgG2c N297A constant heavy chain was then cloned into the pAAV-TRES6 plasmid as described above for the wild-type sequence. Both vector construct sequences have also been deposited at GeneBank (accession numbers PP740716 and PP740719).

### In vitro antibody production, analysis, and purification
The analysis of expression constructs was carried out in six-well cell culture plates (Greiner Bio-One, Frickenhausen, Germany). A total of $10^6$ HEK293T cells were seeded one day before the experiment. On the next day, 3 µg of expression plasmid were transiently transfected by polyethylenimine transfection (PEI; Polysciences, Warrington, PA, USA) using 9 µg PEI. After a 72 h incubation period, the supernatant was harvested and used for further experiments. The binding of secreted antibodies in the cell supernatant was assessed by performing a flow cytometric antibody-binding test, as described previously[67], the gating strategy is shown in Fig. S8. The molecular weight of heavy and light chains was analysed by SDS-PAGE. After gel-electrophoresis, the separated proteins were blotted onto an Amersham™Protran™ 0.45 µm nitrocellulose blotting membrane (Cytiva, Marlborough, MA, USA) using the Pierce Power Blot Cassette system (Thermo Fisher Scientific). After 1 h blocking with 5% skim milk at RT, the membrane was incubated with anti-human IgG1 AlexaFluor488 (Thermo Fisher Scientific) overnight at 4 °C protected from light. On the next day, the membrane was washed, and the fluorescence intensity was measured by the Intas ChemoStar device (Intas Science Imaging Instruments GmbH, Göttingen, Germany).

For the purification of TRES6 antibodies, T175 Greiner flasks (Greiner Bio-One) were seeded with $1.8 \times 10^7$ HEK293T cells and, on the next day transiently transfected with 70 µg plasmid DNA per flask. After 72 h, the cell supernatant was collected, residual cells were pelleted, and the antibody-containing supernatant was filtered through a 0.45 Minisart® syringe filter (Becton Dickinson, Franklin Lakes, NJ, USA). The antibodies were purified using prepacked Protein G GraviTrap™ columns (Cytiva) following the manufacturer's protocol. Bound TRES6 was eluted using 0.1 M glycin buffer (pH 2.7) and subsequently neutralized with 1 M Tris-HCl (pH 9.0). After buffer exchange to PBS and volume reduction using Amicon Ultra centrifugal filters with 30 kDa cut-off (Merck, Darmstadt, Germany), the protein concentration was determined with the Pierce BCA kit (Thermo Fisher Scientific) following instructions of the manufacturer.

### Production of AAV particles
The AAV8 and AAVMYO vector particles were produced by Sirion Biotech (Martinsried, Germany). The production was carried out in ten-layer Cell Factories (CF10) in which $2.5 \times 10^8$ HEK293T cells were seeded per CF10 the day before transfection. Transfections were performed using equimolar

amounts of expression plasmid, helper, and AAV8/AAVMYO packaging plasmids. A total of 2 mg plasmid DNA/CF10 and a PEI-to-DNA ratio of 3 was used. Transfected cells were then incubated for 48 h under standard cell culture conditions (37 °C, 5% CO2). Thereafter, AAV particles in the cell culture medium were precipitated with 8% PEG8000 plus 0.5 M sodium chloride (NaCl) overnight. Harvested cells were resuspended in 100 ml lysis buffer (50 mM Tris, 150 mM NaCl, 5 mM MgCl2, 0.001% poloxamer 188, pH 8.5) per CF10 and frozen at −20 °C. Precipitated AAV particles were centrifuged, resuspended in 5 ml lysis buffer, and frozen at −20 °C. Frozen samples were thawed and pooled.

To harvest cell-associated AAV vector particles, transfected cells were lysed for 1 h at 37 °C in the presence of 0.1% Triton X-100 and benzonase. The lysate was clarified by centrifugation and subsequent filtration (0.45 μm). AAV particles in the clarified lysate were then captured at a flow rate of 1 ml/min on 1 ml Poros CaptureSelect AAVX (Thermo Fisher Scientific) columns using an ÄKTA pure chromatography system (Cytiva). Captured material was eluted with 200 mM citrate buffer at pH 2.0 and collected as 1 ml fractions in tubes containing 850 μl 1 M Tris at pH 8.0 for immediate neutralization. Peak fractions were pooled and stored at 4 °C. For separation of empty and full particles, affinity-captured AAV samples were loaded onto discontinuous iodixanol gradients from 15 to 60% and centrifuged for 3 h at 4 °C and 50,000 rpm using an Optima L-70i ultracentrifuge (both from Beckman Coulter Life Sciences, Brea, CA, USA). Full AAV vectors were then collected from the 40/60% iodixanol interface.

### Formulation and sterile filtration of AAV samples
AAV8 and AAVMYO vector particles were concentrated and formulated using Vivaspin 20 centrifugal concentrators with a 100,000 molecular weight cut-off (Sartorius, Göttingen, Germany). Formulated AAV particles were then sterile-filtered through Acrodisc PP, PES, 0.2 μm, and 1 cm² filter units (Pall Corporation, Port Washington, NY, USA). Two-step-purified particles were formulated in Dulbecco's phosphate-buffered saline (DPBS, 1.47 mM monopotassium phosphate, 2.7 mM potassium chloride, 8.06 mM disodium phosphate heptahydrate, 137 mM NaCl) containing 0.001% (w/v) poloxamer 188. The titers of vector genome-containing AAV particles (vg/ml) were determined by qPCR using ITR-specific primers.

To confirm the purity after production, AAV particles corresponding to $10^{10}$ vg were separated on a 10% SDS-PAGE using Pierce™ Silver stain kit (Thermo Fisher Scientific) for visualization of proteins. Only VP1, VP2, and VP3 proteins in the typical stoichiometry of approximately 1:1:10 were detectable, indicating a purity of the AAV preparation of >95%. Finally, AAV samples were adjusted to a titer of $10^{13}$ vg/ml, aliquoted, and frozen at −80 °C.

### In vitro differentiation, cultivation, and transduction of human iPSC-derived cardiomyocytes
F1 hiPSCs[68] were maintained on Matrigel-coated culture plates, passaged using Accutase in the presence of Y-27632 2HCl (10 μM, Sellekchem) for the first 24 h, and reseeded at passage number 22 to 24 at a density of $2.5 \times 10^4$ cells per well of a Matrigel-coated six-well plate. Cardiac differentiation was performed as described previously[68]. Briefly, more than 80% of confluent hiPSCs were treated with CHIR-99021 (6 μM in RPMI-1640 containing B-27 supplement minus insulin (2%)) for 2 days to induce mesoderm formation by activating Wnt signaling. After 1-day incubation in RPMI-1640 containing B-27 supplement minus insulin (2%), Wnt signalling was inhibited by IWR-1-endo (5 μM in RPMI-1640 containing B-27 supplement minus insulin (2%)) for 2 days. After another 2 days of incubation in a culture medium, cells started to beat. Subsequently, cells were cultured in RPMI-1640 containing B-27 supplement for 2 days, and at day 9 of differentiation, metabolic starvation was induced by substituting glucose with lactate (5 mM in RPMI-1640 minus glucose) for 3 days. After metabolic selection, cardiomyocytes were cultured in RMPI 1640 containing B-27 supplement (2%), and the medium was changed every 2 to 3 days until further experiments were performed.

### Vector application and sample harvesting
AAV8 and AAVMYO vector particles and purified antibodies were applied intravenously by injection into the lateral tail vein. Therefore, male and female TRIANNI mice were placed in a Mouse Restrainer Typ A (G&P Kunststofftechnik, Kassel, Germany) and the tails were pre-warmed for 1 to 2 min under infrared light. Afterward, the samples were injected intravenously using a 30 G insulin syringe (Becton Dickinson) in a total volume of 100 μl PBS.

Blood was collected from the retrobulbar venous plexus using glass capillaries (minicaps®, Hirschmann, Eberstadt, Germany) under isoflurane anesthesia into a Microtainer™ SST tube (Becton Dickinson). The serum was separated by centrifugation for 5 min at 6000×g and stored at −20 °C. At the final time point, the mice were sacrificed by CO2 euthanasia. Prior to organ removal, bronchoalveolar (BAL) fluid was collected. Therefore, a 24 G catheter was inserted into the trachea, the diaphragm was cut open, and the lung was rinsed with 1 ml PBS. Afterward, the fluid was stored at −20 °C. Finally, the organs were removed and stored at −20 °C in RNAlater buffer (Thermo Fisher Scientific).

### Quantification of antibody levels in sera and BAL fluid
The concentration of TRES6 antibodies in sera and BAL fluid of the transduced mice was determined by a quantitative ELISA as described before[36,69]. Briefly, 96-well high-binding assay plates (Corning Inc, Corning, NY, USA) were coated overnight with 100 ng RBD protein (Diarect GmbH, Freiburg, Germany) in 100 μl coating buffer (0.1 M Na2CO3, 0.1 M NaHCO3 in H2O, pH 9.6) per well. After blocking with 5% skimmed milk, serum and BAL samples, appropriately diluted in 2% skimmed milk, were applied. For detection, anti-mouse IgG HRP (1:4000, Dianova, Hamburg, Germany) was added for a 1 h incubation period. For final development, 100 μl of ECL solution was added and the plates were subsequently measured in the Orion microplate luminometer (Berthold Detection Systems GmbH, Pforzheim, Germany). A twofold serial dilution of purified monoclonal TRES6 antibody starting at a concentration of 400 ng/ml was used for the quantitative calculation of the antibody concentration using the GraphPad Prism 6 software performing a point-to-point interpolation (GraphPad Software, Boston, MA, USA). The mean antibody half-life in the TRES6-administered group was calculated from the half-life of serum concentrations of each animal determined at weeks 4, 8, and 12 using the GraphPad Prism 6 software (GraphPad Software) performing a non-linear fit and assuming a one-decay-phase model.

### Analysis of anti-drug antibodies
The presence of ADAs was analysed by performing a bridging ELISA. Twenty-four hours prior to the assay, purified TRES6 was biotinylated using the One-Step Antibody Biotinylation Kit (Miltenyi Biotec, Bergisch Gladbach, Germany) following the manufacturer's instructions. Overnight, 96-well high-binding assay plates (Corning Inc) were coated with 50 ng unlabeled TRES6 antibody in 100 μl coating buffer per well. On the next day, the plates were blocked for 1 h at RT with 5% skimmed milk. Afterward, pre-diluted serum samples (1:80 in 2% skimmed milk) were added, and incubated for 1 h at RT, before the plates were washed three times with PBS containing 0,1% tween (Sigma-Aldrich, St. Louis, MO, USA). Biotinylated TRES6 was added. After a further incubation period for 1 h at RT and a subsequent washing step, streptavidin-HRP (1:1000 in 2% skimmed milk, Abcam, Cambridge, United Kingdom) was added. After additional washing steps, 100 μl of ECL solution was added and the luminescence was subsequently measured in the Orion microplate luminometer (Berthold Detection Systems GmbH). A twofold serial dilution of anti-murine IgG2c (Thermo Fisher Scientific) antibody starting at a concentration of 400 ng/ml was used for the quantitative calculation of the anti-drug-antibody concentration using the GraphPad Prism 6 software performing a point-to-point interpolation (GraphPad Software).

### Pseudotype neutralization assay
A lentiviral pseudotype neutralization assay was performed as described[65,70,71]. Briefly, pseudotyped particles were produced by transient

co-transfection of HEK293T cells with plasmids encoding the wild-type SARS-CoV-2 spike protein with the D614G mutation and lentiviral packaging plasmids[65,72]. The particle-containing supernatant was harvested 72 h after transfection, sterile-filtered through a 0.45 μm Minisart® syringe filter (Becton Dickinson) and a serial dilution was tested for its capacity to transduce HEK293T cells that stably express the human ACE2 enzyme[73]. The dilution resulting in a luciferase signal of 100,000 RLUs applying the One-Glo Luciferase Assay System (Promega, Walldorf, Germany) was selected for further use. To determine the neutralization capacity, a four-fold serial dilution of the sera starting at a 20-fold dilution was incubated for 1 h at 37 °C after the addition of pseudotype particles at a volume ratio of 1:1. Afterward, the mix was added to ACE2-expressing cells that were seeded 1 day before at a density of $5 \times 10^4$ cells per well in 96-well microtiter flat bottom plates (Greiner Bio-One). After 48 h incubation under standard cell culture conditions (37 °C, 5% $CO_2$), the luciferase activity was determined by using the One-Glo Luciferase Assay System (Promega) following the manufacturer's protocol. The emitted signal was recorded in the Multilabel plate reader Victor X4 (Perkin Elmer, Waltham, MA, USA). All values were expressed as percent of the values of wells lacking sera and subtracted from 100% to obtain the percent inhibition. The dilution at which 50% inhibition was reached ($ID_{50}$) was calculated using the GraphPad Prism 9.0.1 software (GraphPad Software) performing a non-linear regression and assuming a sigmoidal 4PL model.

### Quantification of AAV vector DNA and RNA levels

DNA was extracted from collected mouse sera using the GeneJet Genomic DNA purification kit (Thermo Fisher Scientific) and the concentration was measured in the Nanodrop 1000 device (Thermo Fisher Scientific). A volume of 0.5 μl of double-strand-specific DNase (Cat. No. EN0771, Thermo Fisher Scientific) was added to 5 ng extracted DNA. The double-stranded DNase-resistant vector genome copy numbers were quantified by qPCR targeting the CMV promoter (forward primer: 5′-gtcaatgggtggag-tatttacgg-3′; reverse primer: 5′-aggtcatgtactgggcataatgc-3′; probe: 5′-6FAM-caagtgtatcatatgccaagtacgccccc-TAMRA-3′) using the iTaq universal probes Supermix (Bio-Rad Laboratories GmbH, Feldkirchen, Germany) following the manufacturer's instructions. The reaction was measured in the Applied Biosystems™ 7500 Real-Time PCR System (Thermo Fisher Scientific) at these cycling conditions: dsDNA digestion: 5 min at 37 °C, polymerase activation/DNA denaturation/dsDNase inactivation at 95 °C for 5 min followed by 35 amplification cycles consisting of 15 s denaturation at 95 °C and annealing/extension for 60 s at 60 °C. For quantification, a standard curve in the range of $10^{11}$ to $10^2$ vg/ml was used. The calculated copy numbers were ultimately normalized to the serum volume that was initially used for DNA extraction.

Total RNA was extracted from murine organs using the RNAZol RT reagent (Merck) following the protocol given by the manufacturer. The concentration of the total RNA in the extracts was determined using the Nanodrop 1000 device (Thermo Fisher Scientific). The AAV vector RNA in the extracts was detected by RT-qPCR amplifying a sequence of the WPRE (forward primer: 5′-ccgttgtcaggcaacgtg-3′; reverse primer: 5′-agctga-caggtggtggcaat-3′; probe: 5′-6FAM-gctgacgcaaccccccactggt-TAMRA-3′). The 20 μl reaction mix consisted of 2 μl RNA sample at a concentration of 5 ng/μl, 400 nM of each primer, 200 nM probe and the Luna Universal Probe One-Step RT-qPCR reagents (New England BioLabs GmbH). Additionally, 0.5 μl dsDNase (Cat. No. EN0771, Thermo Fisher Scientific) was added in order to degrade residual viral DNA and the samples were transferred into a Sapphire 96-well microplate (Greiner Bio-One GmbH).

Measurement was carried out in the Applied Biosystems™ 7500 Real-Time PCR System (Thermo Fisher Scientific) using the following cycle conditions: dsDNA digestion: 5 min at 37 °C, reverse transcription/dsDNase denaturation: 10 min at 55 °C, initial denaturation: 1 min at 95 °C, 35 cycles: denaturation: 10 s at 95 °C, extension: 60 s at 60 °C. Acquired data were analysed using the 7500 Software (Thermo Fisher Scientific). For quantification, a standard curve in the range of $10^{11}$ to $10^2$ vg/ml was used.

The calculated sample concentration was ultimately normalized to the RNA input in ng.

The complete enzymatic removal of residual AAV vector DNA in the RNA samples was additionally verified by setting up a control qPCR reaction targeting the WPRE for each sample using the iTaq universal probes Supermix (Bio-Rad Laboratories GmbH). Here, the 20 μl reaction mix consisted as well of a 2 μl RNA sample at a concentration of 5 ng/μl, 400 nM of each primer, 200 nM probe and the iTaq universal probes Supermix (Bio-Rad Laboratories GmbH), and 0.5 μl dsDNase (Cat. No. EN0771, Thermo Fisher Scientific). The measurement was carried out as described above for the quantification of vector DNA in the sera.

### Anti-drug T cell responses

Splenocytes of transduced mice were seeded after red blood cell lysis in ACK buffer (150 mM $NH_4Cl$, 10 mM $KHCO_3$, 0.1 mM EDTA in $H_2O$, pH at 7.4) at a density of $10^6$ cells per well in a 96-well round bottom cell culture plate (Greiner Bio-one). Aliquots were stimulated separately with different TRES6-derived peptides at a final concentration of 10 μg/ml (peptide sequences: CDR1 HC: GFTFSSYGMHWVRQA-$NH_2$, CDR2 HC: VIWYD GSNKYYADSVKG-$NH_2$, CDR3 HC: YCVRETVDGMDVWGQ-$NH_2$, Furin/F2A: RSGSGAPVKTLNFDL-$NH_2$, F2A: VKTLNFDLLKLAGD-$NH_2$, F2A/LC: LKLAGDVESNPGPMD-$NH_2$, CDR1 LC: TCRARQDINN YLAWF-$NH_2$, CDR2 LC: HLIYAASSLLSGVPS-$NH_2$, CDR3 LC: YYCLQH NSYPYTFGQ-$NH_2$) in the presence of 2 μg/ml anti-mouse CD28 (eBioscience™, Thermo Fisher Scientific) and 3 μg/ml Brefeldin A (Thermo Fisher Scientific). The cells were restimulated overnight under standard cell culture conditions. On the next day, the cells were stained for surface markers with anti-mouse CD4 Super Bright 600 (1:300, clone RM4-5, Thermo Fisher Scientific), anti-mouse CD8 FITC (1:300, clone 53-6.7, BD Bioscience) and anti-mouse CD3 Alexa Fluor 700 (1:300, clone 17A2, Bio-legend, San Diego, CA, USA) after previous Fc-blocking using anti-mouse CD16/CD32 antibodies (1:300, clone 93, eBioscience™, Thermo Fisher Scientific). For the identification of dead cells, Fixable Viability Dye eFluor450 (1:400, Thermo Fisher Scientific) was included. After paraformaldehyde fixation (2%, 20 min at RT; Carl Roth, Karlsruhe, Germany) and saponin permeabilization (0.5%, 30 min at 4 °C, Sigma-Aldrich), the intracellular cytokines were stained using anti-mouse IL-2 APC (1:300, clone JES6-5H4), anti-mouse IFN Phycoerythrin (1:300, clone XMG1.2) and anti-mouse TNF PE-Cy7 (1:300, cloneMP6-XT22; all eBioscience™, Thermo Fisher Scientific). The flow cytometric measurement was carried out using the Attune NxT device (Thermo Fisher Scientific), the collected data were analysed with FlowJo (BD Bioscience) and statistical evaluation was performed with the GraphPad Prism 9.0.1 software (GraphPad Software). The gating strategy id shown in Fig. S9.

### Fc-γ receptor binding ELISA

In order to analyse the binding of in vivo produced TRES6 in the murine sera to Fc-γ receptors, 96-well high-binding assay plates (Corning Inc) were coated with 100 ng tag-free RBD protein (Genaxxon Bioscience, Ulm, Germany) per well in coating buffer overnight. On the next day, the plates were washed and blocked for 1 h at RT with 5% skimmed milk. The serum samples were adjusted to a TRES6 concentration of 100 ng/ml. After washing the RBD-coated plates three times, PBS containing 0,1% tween (Sigma-Aldrich), the diluted sera were added for an incubation period of 1 h at RT. After repeating the washing steps, 50 ng of His-tagged recombinant murine Fc-γ RI, RIIb, RIII (all R&D Systems, Minneapolis, MN, USA), or RIV (Abeomics, San Diego, CA, USA) were added and incubated for 1 h at 37 °C. For detection, the plates were afterward washed three times with PBS containing 0,1% tween (Sigma-Aldrich), anti-6-His HRP antibody (1:1000, Bethyl Laboratories, Waltham, MA, USA) was applied for 1 h at RT, and the plates were developed after thorough washing by addition of 100 μl ECL solution. The measurement was carried out using the Multilabel plate reader Victor X4 (Perkin Elmer) and the data were statistically analysed with GraphPad Prism 6 software (GraphPad Software).

## Anti-spike (S) murine IgG2c Fc-glycosylation analysis

Anti-S IgG Fc-glycosylation analysis was performed as described previously[74]. In short, anti-S IgGs were affinity-purified using solid phase-adsorption on microtiter plates (Thermo Fisher Scientific) coated with SARS-CoV-2 S trimer that were produced as described in ref. [75]. Antibodies were eluted from the plates with 100 mM formic acid, subjected to trypsinization, and resulting IgG Fc glycopeptides were registered by liquid chromatography-mass spectrometry. Glycan compositions were assigned on the basis of accurate mass, retention time, and literature[76], supported by collision-induced dissociation tandem mass spectra. Relative quantification of glycoforms was performed on the basis of mass spectrometry signal intensities, and glycosylation traits including fucosylation, galactosylation, and sialylation were calculated.

## Murine challenge experiment

Transgenic K18-hACE2 mice were obtained from a commercial supplier (Jackson Laboratory, Bar Harbor, ME, USA) and bred at Fraunhofer Institute for Cell Therapy and Immunology (IZI). Female mice were randomly assigned into groups of six animals and kept under standard conditions in isolated ventilated cages. Mice were intravenously injected with either $5 \times 10^{11}$ vg AAV8-TRES6 or $5 \times 10^{11}$ vg AAVMYO-TRES6 14 days prior to virus inoculation or 3.33 mg/kg TRES6 N297A, 3.33 mg/kg TRES6 or 3.33 mg/kg isotype control antibodies 5 days prior to virus inoculation. Mice were infected intranasally under isoflurane anesthesia with 300 FFU of SARS-CoV-2 (Wuhan strain) in 50 µl total volume. Animals were monitored daily for body weight and clinical score. The following parameters were evaluated in the scoring system: weight loss and body posture (0–20 points), general conditions including the appearance of fur and eye closure (0–20 points), reduced activity and general behavior changes (0–20 points), and limb paralysis (0–20 points). Cohort 1 ($n = 6$) was euthanized at day 4 and cohort 2 ($n = 6$) was euthanized according to humane endpoints (latest at day 10) if a cumulative clinical score of 20 or more was reached. Indicated organs were homogenized in 2 ml ice-cold PBS after collection using the gentleMACS Octo Dissociator (Miltenyi Biotec). Afterward, tissue homogenates were centrifuged at 2000×g for 5 min at 4 °C to separate cell debris and the supernatant was removed and stored at −80 °C until viral RNA isolation. Viral RNA was isolated from 140 µl of homogenates using the QIAamp Viral RNA Mini Kit (Qiagen, Hilden Germany). RT-qPCR reactions were performed using TaqMan® Fast Virus 1-Step Master Mix (Thermo Fisher Scientific) and 5 µl of isolated RNA as a template to detect a 132 bp sequence in the ORF1b/NSP14 as previously described[36]. Tenfold serial dilutions of synthetic SARS-CoV-2-RNA (Twist Bioscience, San Francisco, CA, USA) in the range of $5 \times 10^6$ to $5 \times 10^2$ copies/5 µl were used as a quantitative standard to obtain viral copy numbers.

## Transmission electron microscopy

Negative staining transmission electron microscopy was performed as described before[77] with the following changes. Samples were stained using a 2% uranyl acetate solution and, after air-drying, transferred to a JEOL1400 Plus transmission electron microscope (JEOL, Munich, Germany). Images were acquired at a nominal magnification of 30,000[78].

## Data availability

All reported data of the main text can be accessed in the Supplementary Data 1 file.

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

## Acknowledgements

This work was supported by grants from the Bayerische Forschungsstiftung (AZ-1458-20) and the Deutsche Forschungsgemeinschaft (Ue45/23-1) and Projektnummer 326998133 –TRR 225 (subproject C01 to F.B.E.). We would like to thank Hans-Martin Jäck for providing breeding pairs of TRIANNI mice, Anne-Kathrin Donner and Isabell Schulz for excellent technical assistance, and the group of Dr. Jasmin Fertey (Fraunhofer Institute for Cell Therapy and Immunology (IZI), Leipzig, Germany) for providing viral stocks for mouse infections. We also would like to thank Dr. Ronald C. Desrosiers and Dr. Sebastian Fuchs for providing detailed sequence information on their antibody-encoding AAV vector construct.

## Author contributions

Conceptualization, K.Ü., C.T., T.G., and M.W.; Methodology and investigation, J.W., S.M.M.-S., W.W., P.A., N.U., L.I., V.E., D.D., K.R., A.E., F.O., A.S.P., V.T., S.S., F.B.E., T.G., M.W.; Writing original draft, J.W. and K.Ü. Writing review and editing, all authors. Resources, D.G.; Funding acquisition: K.Ü., F.B.E., and C.T.

## Funding

## Competing interests

D.G. has filed a patent application on the myotropic AAV capsid; S.S. and C.T. are employees of the company Revvity Gene Delivery GmbH offering vector production services. The remaining authors declare no competing interests.
