## [Peer Review File · Communications Biology]

Reviewers' comments:

Reviewer #1 (Remarks to the Author):

The manuscript compared an AAV8 hepatotropic and an AAVMYO myotropic vector for SARS-CoV-2 neutralizing antibody (TRES6) delivery and protection against SARS-CoV-2 infection in K18-hACE2 mice. The study showed that the myotropic AAV capsids could produce a higher serum level of TRES6 and maintained above 100 µg/ml for one year, as well as favourable Fc-γ receptor binding activities. Overall, the work presented in the manuscript is very interesting and may provide another potential tool in the fight against COVID infection. I have a couple of minor comments for the authors' consideration.

1. Half-life of TRES6 antibody in TRIANNI mice: The authors stated that the half-life of TRES6 antibody in TRIANNI mice was 7 days (Line 238). However, Figure 5A showed that following a single dose of 5 mg/kg TRES6 antibody, it took 24 weeks to eliminate the antibody. The half-life of TRES6 appears to be 4-5 weeks. Please clarify how the half-life of 7 days was derived.
2. Figure 5B: Please add the unit of AUC.
3. Figure 6B: AAV vector genome is usually rapidly eliminated following IV administration. Please explain why AAV vector RNA levels in liver and muscles increased from Week 4 to Week 52.
4. Peak serum concentrations of the AAVMYO-delivered antibody could reach serum levels in an estimated range of 5 % of the total IgG concentration in human serum. Is there any literature report on the highest TRES6 level tolerated in mice and human? Is there any safety concern with TRES6 > 500 µg/mL?

Reviewer #2 (Remarks to the Author):

For this manuscript the authors constructed optimized AAV vectors carrying the human variable portion of TRES6 SARS-CoV-2 neutralizing Ab fused to a murine Ig constant region packaged in either hepatotropic (AAV8) or myotropic capsids, then administered them to TRIANNI mice and defined the antibody PKs. The author's found that the myotropic vector produced the highest level of TRES6 Ab with serum levels more than 100µg/ml for 1 year. After 1 year, levels of TRES6 Ab were higher in animals given the vector packaged in a myotropic capsid. During this period no anti-drug antibodies were identified in either animal groups. Expression of vector mRNA appeared to be stable between 4 and 52 weeks in target organs. Vector delivered in hepatotropic capsids had detectable viral RNA in hepatocytes, vector delivered in myotropic capsids had detectable viral RNA in heart and quadricep muscle. Administration of TRES6 packaged in AAV8 or myotropic capsids 14 days prior to challenge with 300FFU Wuhan strain SARS-CoV-2 resulted in similar levels of protection (100%) in these mice. Other clinical parameters were also similar. The authors found the glycosylation pattern on the TRES6 antibodies were different in antibodies produced in liver compared to those produced in muscle and heart with higher levels of fucosylation found in antibodies produced in heart and muscle. The authors further show differences in murine Fc receptor binding to in vivo produced hepatotropic and myotropic antibodies and suggest that these differences may be responsible for increased levels of TRES6 antibodies in animals injected with

myotropic vectors.

This manuscript represents a carefully done work that addresses potentially significant medical issues, increasing the expression AAV vector products and minimizing possible inflammatory risks of product administration. In making their argument the authors administered 5×10^{11} vg/mouse of AAV8-TRES6, AAV8-TRES6 N297A and AAVMYO-TRES6. Although all 3 vectors yield good TRES6 levels only the AAVMYO-TRES6 continued to increase after week 3 with a four fold difference in concentration at 52 wks. Consider the difference in capsids, it is possible that the myotropic and hepatotropic vectors are not cleared from the blood at the same rate. This may give one of the two vectors a greater time to infect cells. The authors do not consider this possibility. This can easily be determined using qPCR with primers and probes crossing the 3' HC region and the F2A region.

Below are specific points that should be addressed before acceptance:

1. The clearance rate of AAV8 and myotropic vectors from serum should be determined.
2. The authors indicate that the sex of the mice effects fucosylation of the HC, with female mice having higher levels of fucosylation. In the methods section the authors state that for the murine challenge experiment, only female mice were used. What happens when male mice are used? At a minimum for each figure the authors should state the number of male and female mice in each group.

Lesser points:

1. From the author's introduction it seems that all antibodies produced by AAV vectors in humans are cleared from blood by ADA. That is not true, human studies have shown the persistence of VRC07 in serum at levels $>1\mu\text{g}$ for over a year in the absence of measurable ADA.
2. The production of antibodies from organs which do not usually produce antibodies raises the risk of an autoimmune reaction against that organ. Antibody-mediated autoimmune reactions directed against cardiac myocytes are known. The authors should mention these concerns in the discussion. Fucosylation of the Fc region may decrease this risk.

Reviewer #3 (Remarks to the Author):

COMMSBIO-23-4356-T Review

Key results: The manuscript by Wagner et al., entitled "Influence of AAV vector tropism on long-term expression and Fc- γ receptor binding of an antibody targeting SARS-CoV-2" evaluated AAV-mediated expression of the anti-SARS-CoV-2 mAb, TRES6, in a murine model. The authors monitored AAV-mAb expression kinetics, glycosylation, anti-drug antibody (ADA) responses to TRES6 expressed from two different AAV capsids: liver tropic AAV8 and muscle tropic AAVMYO, following IV administration to mice. Analysis of the Fc-glycosylation patterns of AAV-expressed TRES6 mAb revealed critical differences between the two capsids in terms of the mAb's binding activities to murine Fc- γ receptors. The authors conclude that use of a muscle tropic AAV capsid, such as AAVMYO, leads to more robust and long term mAb expression, lack of ADA, and more favourable Fc- γ receptor interactions compared to TRES6 expressed from the liver tropic AAV8. Finally, the authors demonstrate that AAV-mediated expression of the TRES6 mAb protected K18-hACE2 mice from challenge with ancestral SARS-CoV-2. The manuscript is well written and informative and provides new insights into mAb glycosylation patterns when expressed de novo

following AAV mAb gene delivery.

Originality and significance: The comparison between to the two AAV capsids and the impact on glycosylation is novel and will be of interest to those in the field of AAV vectored immunoprophylaxis as well as those with a general interest in AAV gene therapy.

Data & methodology: The methodology described is sound. The experiments were rigorously conducted with the appropriate controls.

Appropriate use of statistics and treatment of uncertainties: All error bars are defined in the corresponding figure legends and the statistical tests used were appropriate.

Conclusions: The conclusions and data interpretation are robust, valid, and reliable.

References:

- Reference 22 and 23 do not appear to be appropriate as they do not discuss the use of AAV to express antiviral mAbs. Please revise.
- Line 79: Rghei et al showed mAb expression for close to three years in sheep without the development of ADA. See <https://doi.org/10.1038/s41434-022-00361-2>
- Line 81: AAV VIP studies in NHP was recently summarized in DOI: 10.3390/biomedicines11082223

Clarity and context: The abstract is clear and accessible. The abstract, introduction and conclusions are appropriate.

Suggested improvements:

1. Line 49: The abstract refers to the mAb TRES6 being fused to a murine constant domain, but Figure 1 indicates that a human IgG1 heavy chain constant domain was used. Since the authors evaluated three different versions of TRES6, including a murine, a human, and a human version with the heavy and light chain reversed, it would be helpful to include an image of all three mAbs in Figure 1.
2. Line 153: What was the rationale for changing to a mouse Fc domain? There are numerous reports in the literature of AAV VIP studies in mice (as well as guinea pigs, hamsters and sheep) where human IgG1 mAbs have been efficiently expressed from AAV, in some cases for the lifetime of the animal.
3. 197: Add citation to support this statement.
4. Figure 4: Did the authors attempt to quantify the amount of replication competent SARS-CoV-2 in the lungs? No TCID50 data is presented.
5. Line 277: Revise as follows: “Given previous reports on frequent induction of ADAs after AAV-based delivery of antibodies in NHPs and humans...”

6. Line 310: The sentence appears to be incomplete.

Influence of AAV vector tropism on long-term expression and Fc- γ receptor binding of an antibody targeting SARS-CoV-2

Wagner JT^{1*}, Müller-Schmucker SM¹, Wang W², Arnold P³, Uhlig N⁴, Issmail L⁴, Eberlein V⁴, Damm D¹, Roshanbinfar K⁵, Ensser A¹, Oltmanns F¹, Peter AS¹, Temchura V¹, Schrödel S⁶, Engel FB⁵, Thirion C⁶, Grunwald T⁴, Wuhrer M², Grimm D⁷, Überla K^{1*}

Point-to-point response to reviewers:

Reviewer #1 (Remarks to the Author):

The manuscript compared an AAV8 hepatotropic and an AAVMYO myotropic vector for SARS-CoV-2 neutralizing antibody (TRES6) delivery and protection against SARS-CoV-2 infection in K18-hACE2 mice. The study showed that the myotropic AAV capsids could produce a higher serum level of TRES6 and maintained above 100 $\mu\text{g/ml}$ for one year, as well as favourable Fc- γ receptor binding activities. Overall, the work presented in the manuscript is very interesting and may provide another potential tool in the fight against COVID infection. I have a couple of minor comments for the authors' consideration.

1. Half-life of TRES6 antibody in TRIANNI mice: The authors stated that the half-life of TRE6 antibody in TRIANNI mice was 7 days (Line 238). However, Figure 5A showed that following a single dose of 5 mg/kg TRES6 antibody, it took 24 weeks to eliminate the antibody. The half-life of TRES6 appears to be 4-5 weeks. Please clarify how the half-life of 7 days was derived.

The half-life of the antibody was determined from week 4 to week 12 data and the precise calculated half-life is now given the revised manuscript (Line 247).

A short description on the calculation of the half-life is also provided in the revised manuscript in the method section. (Line 650 - 653).

2. Figure 5B: Please add the unit of AUC.

Changed as suggested.

3. Figure 6B: AAV vector genome is usually rapidly eliminated following IV administration. Please explain why AAV vector RNA levels in liver and muscles increased from Week 4 to Week 52.

A hypothesis for the observed increase of vector RNA is now included in the discussion section of the revised manuscript. (Line 428 - 435)

4. Peak serum concentrations of the AAVMYO-delivered antibody could reach serum levels in an estimated range of 5 % of the total IgG concentration in human serum. Is there any literature report on the highest TRES6 level tolerated in mice and human? Is there any safety concern with TRES6 > 500 $\mu\text{g/mL}$?

A discussion on the safety of TRES6 and antiviral monoclonal antibodies is now provided in the revised manuscript. (Line 384 - 388)

Reviewer #2 (Remarks to the Author):

For this manuscript the authors constructed optimized AAV vectors carrying the human variable portion of TRES6 SARS-CoV-2 neutralizing Ab fused to a murine Ig constant region packaged in either hepatotropic (AAV8) or myotropic capsids, then administered them to TRIANNI mice and defined the antibody PKs. The author's found that the myotropic vector produced the highest level of TRES6 Ab with serum levels more than 100ug/ml for 1 year. After 1 year, levels of TRES6 Ab were higher in animals given the vector packaged in a myotropic capsid. During this period no anti-drug antibodies were identified in either animal groups. Expression of vector mRNA appeared to be stable between 4 and 52 weeks in target organs. Vector delivered in hepatotropic capsids had detectable viral RNA in hepatocytes, vector delivered in myotropic capsids had detectable viral RNA in heart and quadricep muscle. Administration of TRES6 packaged in AAV8 or myotropic capsids 14 days prior to challenge with 300FFU Wuhan strain SARS-CoV-2 resulted in similar levels of protection (100%) in these mice. Other clinical parameters were also similar. The authors found the glycosylation pattern on the TRES6 antibodies were different in antibodies produced in liver compared to those produced in muscle and heart with higher levels of fucosylation found in antibodies produced in heart and muscle. The authors further show differences in murine Fc receptor binding to in vivo produced hepatotropic and myotropic antibodies and suggest that these differences may be responsible for increased levels of TRES6 antibodies in animals injected with myotropic vectors. This manuscript represents a carefully done work that addresses potentially significant medical issues, increasing the expression AAV vector products and minimizing possible inflammatory risks of product administration. In making their argument the authors administered 5 x 10¹¹ vg/mouse of AAV8-TRES6, AAV8-TRES6 N297A and AAVMYO-TRES6. Although all 3 vectors yield good TRES6 levels only the AAVMYO-TRES6 continued to increase after week 3 with a four fold difference in concentration at 52 wks. Consider the difference in capsids, it is possible that the myotropic and hepatotropic vectors are not cleared from the blood at the same rate. This may give one of the two vectors a greater time to infect cells. The authors do not consider this possibility. This can easily be determined using qPCR with primers and probes crossing the 3' HC region and the F2A region.

Below are specific points that should be addressed before acceptance:

1. The clearance rate of AAV8 and myotropic vectors from serum should be determined.

This aspect is now addressed in the revised manuscript. (Line 307 - 316)

2. The authors indicate that the sex of the mice effects fucosylation of the HC, with female mice having higher levels of fucosylation. In the methods section the authors state that for the murine challenge experiment, only female mice were used. What happens when male mice are used? At a minimum for each figure the authors should state the number of male and female mice in each group.

We have not used male mice. As suggested by the reviewer, we now provide the number of female animals included in each group of the challenge experiment. (see Figure legends)

Lesser points:

1. From the author's introduction it seems that all antibodies produced by AAV vectors in humans are cleared from blood by ADA. That is not true, human studies have shown the persistence of VRC07 in serum at levels >1ug for over a year in the absence of measurable ADA.

The introductory section of the manuscript was rephrased to avoid the impression that ADA leads to clearance of administered antibodies in all participants. (Line 80)

2. The production of antibodies from organs which do not usually produce antibodies raises the risk of an autoimmune reaction against that organ. Antibody-mediated autoimmune reactions directed against cardiac myocytes are known. The authors should mention these concerns in the discussion. Fucosylation of the Fc region may decrease this risk.

This aspect is now discussed in the revised manuscript (Line 388- 394)

Reviewer #3 (Remarks to the Author):

COMMSBIO-23-4356-T Review

Key results: The manuscript by Wagner et al., entitled “Influence of AAV vector tropism on long-term expression and Fc- γ receptor binding of an antibody targeting SARS-CoV-2” evaluated AAV-mediated expression of the anti-SARS-CoV-2 mAb, TRES6, in a murine model. The authors monitored AAV-mAb expression kinetics, glycosylation, anti-drug antibody (ADA) responses to TRES6 expressed from two different AAV capsids: liver tropic AAV8 and muscle tropic AAVMYO, following IV administration to mice. Analysis of the Fc-glycosylation patterns of AAV-expressed TRES6 mAb revealed critical differences between the two capsids in terms of the mAb’s binding activities to murine Fc- γ receptors. The authors conclude that use of a muscle tropic AAV capsid, such as AAVMYO, leads to more robust and long term mAb expression, lack of ADA, and more favourable Fc- γ receptor interactions compared to TRES6 expressed from the liver tropic AAV8. Finally, the authors demonstrate that AAV-mediated expression of the TRES6 mAb protected K18-hACE2 mice from challenge with ancestral SARS-CoV-2. The manuscript is well written and informative and provides new insights into mAb glycosylation patterns when expressed de novo following AAV mAb gene delivery.

Originality and significance: The comparison between the two AAV capsids and the impact on glycosylation is novel and will be of interest to those in the field of AAV vectored immunoprophylaxis as well as those with a general interest in AAV gene therapy.

Data & methodology: The methodology described is sound. The experiments were rigorously conducted with the appropriate controls.

Appropriate use of statistics and treatment of uncertainties: All error bars are defined in the corresponding figure legends and the statistical tests used were appropriate.

Conclusions: The conclusions and data interpretation are robust, valid, and reliable.

References:

- Reference 22 and 23 do not appear to be appropriate as they do not discuss the use of AAV to express antiviral mAbs. Please revise.

Revised as suggested by the reviewer. (Line 74 - 75)

- Line 79: Rghei et al showed mAb expression for close to three years in sheep without the development of ADA. See <https://doi.org/10.1038/s41434-022-00361-2>

The introductory section was rephrased now also providing examples for long-term mAb expression in animal models. (Line 78 - 80)

- Line 81: AAV VIP studies in NHP was recently summarized in DOI: 10.3390/biomedicines11082223

The review on AAV VIP is included in the references of the revised manuscript. (Line 82)

Clarity and context: The abstract is clear and accessible. The abstract, introduction and conclusions are appropriate.

Suggested improvements:

1. Line 49: The abstract refers to the mAb TRES6 being fused to a murine constant domain, but Figure 1 indicates that a human IgG1 heavy chain constant domain was used. Since the authors evaluated three different versions of TRES6, including a murine, a human, and a human version with the heavy and light chain reversed, it would be helpful to include an image of all three mAbs in Figure 1.

As suggested by the reviewer, maps of all three vector constructs are now shown in Figure 1 of the revised manuscript.

2. Line 153: What was the rationale for changing to a mouse Fc domain? There are numerous reports in the literature of AAV VIP studies in mice (as well as guinea pigs, hamsters and sheep) where human IgG1 mAbs have been efficiently expressed from AAV, in some cases for the lifetime of the animal.

A rationale for changing to the mouse Fc domain is provided in the revised manuscript. (Line 156 - 166)

3. 197: Add citation to support this statement.

A citation was added as requested. (Line 206 - 207)

4. Figure 4: Did the authors attempt to quantify the amount of replication competent SARS-CoV-2 in the lungs? No TCID50 data is presented.

Since the administered antibodies can interfere with the quantification of replication competent SARS-CoV-2 from tissues of the animals, TCID50 values were not determined.

5. Line 277: Revise as follows: "Given previous reports on frequent induction of ADAs after AAV-based delivery of antibodies in NHPs and humans..."

The section was revised as suggested by the reviewer. (Line 286 - 287)

6. Line 310: The sentence appears to be incomplete.

The sentence was rephrased. (Line 329 - 332)

REVIEWERS' COMMENTS:

Reviewer #1 (Remarks to the Author):

The authors have addressed my comments. I have no further comments. I recommend acceptance of this manuscript for publication.

Reviewer #3 (Remarks to the Author):

The authors have addressed all my questions and concerns.

I have also looked at authors' responses to Reviewer 2's comments. In my opinion, the authors have adequately addressed Reviewer #2's questions and this has greatly strengthened the manuscript.